# MutS functions as a clamp loader by positioning MutL on the DNA during mismatch repair

Xiao-Wen Yang [ORCID][1,4], Xiao-Peng Han[1,4], Chong Han[1], James London[2], Richard Fishel[2,3] ✉ & Jiaquan Liu [ORCID][1] ✉

Highly conserved MutS and MutL homologs operate as protein dimers in mismatch repair (MMR). MutS recognizes mismatched nucleotides forming ATP-bound sliding clamps, which subsequently load MutL sliding clamps that coordinate MMR excision. Several MMR models envision static MutS-MutL complexes bound to mismatched DNA via a positively charged cleft (PCC) located on the MutL N-terminal domains (NTD). We show MutL-DNA binding is undetectable in physiological conditions. Instead, MutS sliding clamps exploit the PCC to position a MutL NTD on the DNA backbone, likely enabling diffusion-mediated wrapping of the remaining MutL domains around the DNA. The resulting MutL sliding clamp enhances MutH endonuclease and UvrD helicase activities on the DNA, which also engage the PCC during strand-specific incision/excision. These MutS clamp-loader progressions are significantly different from the replication clamp-loaders that attach the polymerase processivity factors β-clamp/PCNA to DNA, highlighting the breadth of mechanisms for stably linking crucial genome maintenance proteins onto DNA.

Mismatch repair (MMR) is an excision-resynthesis process that principally corrects replication errors which produce mismatched nucleotides or insertion-deletion loops in the DNA[1–4]. Defects in MMR genes increase cellular mutation rates more than 100-fold and are the cause of the common cancer predisposition Lynch syndrome or hereditary non-polyposis colorectal cancer[5]. MMR components have also been linked to DNA damage signaling[6] and modulation of cancer immunotherapy[7,8]. MutS homologs (MSH) and MutL homologs (MLH/PMS) are conserved across biology[1–4]. Both the MSH and MLH/PMS proteins bind and hydrolyze ATP, and are responsible for organizing downstream MMR excision events that ultimately remove the error containing strand[9–14].

*E.coli* MMR begins with a mismatch search by a MutS protein homodimer that contains classic Walker A/B ATP binding domains[15,16]. Mispair recognition triggers ATP binding and the formation of a sliding clamp that dissociates from the mismatch and freely diffuses along the adjacent DNA[15,17–21]. Protein structures and single-molecule imaging showed that the MutS sliding clamp recruits a MutL protein homodimer onto the DNA by initially forming a complex between MutS and a MutL N-terminal domain containing the GHKL-superfamily (*G*yrase, *H*sp90, histidine *k*inase and Mut*L*) ATPase[13,22,23]. ATP binding-dependent dimerization of the MutL N-terminal domains (NTDs)[24] leads to the formation of a ring-like MutL clamp on the mismatched DNA (Supplementary Fig. 1a)[13]. The MutL clamp may dissociate from

[1]State Key Laboratory of Molecular Biology, Shanghai Key Laboratory of Molecular Andrology, CAS Center for Excellence in Molecular Cell Science, Shanghai Institute of Biochemistry and Cell Biology, University of Chinese Academy of Sciences, Chinese Academy of Sciences, 320 Yueyang Road, Shanghai 200031, China. [2]Department of Cancer Biology and Genetics, The Ohio State University Wexner Medical Center, Columbus, OH 43210, USA. [3]The Molecular Carcinogenesis and Chemoprevention Program, The James Comprehensive Cancer Center, Columbus, OH 43210, USA. [4]These authors contributed equally: Xiao-Wen Yang, Xiao-Peng Han. ✉e-mail: rfishel@osu.edu; liujiaquan@sibcb.ac.cn

MutS and slide freely along the mismatch DNA and/or oscillate as a MutS-MutL sliding clamp complex with altered diffusion properties[13]. A similar cascade of sliding clamp progression has been observed with the human MSH2-MSH6 and MLH1-PMS2 heterodimers[25].

The MLH/PMS proteins mediate multiple protein-protein interactions to connect mismatch recognition with precise strand-specific excision[26]. In *E.coli* where DNA adenine methylation (Dam) is utilized to discriminate between parental and the error-containing strand, the MutL sliding clamp engages the MutH endonuclease to introduce strand breaks into hemimethylated GATC sites that only transiently occur following replication[13,27]. These long-lived MutL sliding clamps also capture the UvrD helicase and act as a processivity factor in the displacement of the error-containing strand[13,14]. MutH is not conserved outside of a small number of γ-proteobacteria family members that includes *E.coli*[28]. Thus, the detailed strand discrimination signals and excision mechanics remain under intense investigation in most organisms including higher eukaryotes[1,29,30].

MSH and MLH/PMS proteins are not the only ring-shaped molecules that are linked to the DNA. DNA replication relies on a structurally conserved sliding clamp, β-clamp in prokaryotes and PCNA in eukaryotes, which is loaded onto a primer-template by a multiprotein clamp loader complex[31]. The β-clamp/PCNA principally functions as a processivity factor that tethers the polymerase to the DNA as well as a platform to exchange bypass repair polymerases and other DNA metabolic proteins during replication[32,33]. There appears to be several mechanisms utilized by replication clamp loaders for attaching a β-clamp/PCNA to a primer-template junction[32,34]. Yet, all these clamp loaders commonly form an ATP binding-dependent solution complex with the multimeric-ring of β-clamp/PCNA, with the clamp loader eventually transferring the β-clamp/PCNA ring to the DNA utilizing ATP hydrolysis[32,34]. The available evidence suggests that the replication clamp loaders exploit the ATPase cycle to open and close the β-clamp/PCNA ring during DNA loading.

The MMR-dependent clamp loading progressions that lead to the formation of MLH/PMS sliding clamps are largely unknown. The MLH/PMS proteins contain three domains that include the NTD GHKL-ATPase domain, a C-terminal domain (CTD) that stably links protein subunits[24,35–39] and a flexible disordered linker region that connects the NTD and CTD[25] (Supplementary Fig. 1a). The disordered linker has been suggested to compact during ATP binding[40]. Such an MLH/PMS conformational condensation has been proposed to foster the formation of a static complex with MSH proteins at or near the mismatch, capable of capturing a looped DNA or facilitating MLH/PMS polymerization to activate downstream MMR excision components[3,41–43]. In support of these schemes, a positively charged cleft (PCC) was identified in the MLH/PMS NTD domains that was connected to an intrinsic DNA binding activity detected under very low ionic strength conditions[23,24,37,44–48]. In addition, mutations of several PCC residues resulted in impaired MMR[23,37,44,47–49]. However, single-molecule analysis has suggested that MLH/PMS proteins do not stably bind DNA at physiological ionic strength[13,50] and both MSH and MLH/PMS proteins remain continuously dynamic during MMR[13,14,25,27]. Thus, the functional role of the MutL PCC in the multi-step process of MMR remains enigmatic.

Here we have employed single-molecule fluorescence imaging to probe the clamp loader mechanics of *E. coli* MutS with MutL. We show that the efficient formation of a MutL sliding clamp requires a concurrent interaction between the MutL protein, a MutS sliding clamp, and the DNA. Conserved Arg/Lys residues within the MutL PCC appear to provide a crucial DNA docking environment that is exploited by the MutS sliding clamp during MutL clamp-loader functions. The MutL PCC is also employed by the MutL-MutH complex to search and incise hemimethylated GATC sites as well as the UvrD helicase during the capture and unwinding of the mismatched DNA from a strand break. The MutS clamp loader progressions are significantly different from

the replication clamp loaders and expand the repertoire of clamps and clamp loading mechanism utilized for essential genome maintenance DNA transactions.

## Results

### The MutL NTD positively charged cleft is required for MMR

The NTD of MLH/PMS proteins includes the essential GHKL ATPase residues, which appear to fold into an active conformer containing a surface PCC (Fig. 1a, left). The PCC is comprised of embedded Arg/Lys residues that are largely conserved across species (Fig. 1a, right, blue). Three of those conserved residues in the *E.coli* MutL, R162, R266 and R316, have been previously shown to slow the dissociation kinetics of ATP-bound MutS sliding clamps from mismatched DNA, and, when altered increase the cellular mutation rate similar to *E.coli* containing a deletion of MutL (Δ*mutL*) (Fig. 1a, green arrowheads and black boxes)[24,49].

We employed prism-based single-molecule total internal reflection fluorescence (smTIRF) microscopy to visualize activities controlled by the *E.coli* MutL PCC in real time[13,14,25]. Single 18.4-kb DNA molecules containing a G/T mismatch were stretched across a passivated custom-made flow cell surface by laminar flow and linked at both ends via biotin-neutravidin (64% of full length; Supplementary Fig. 2a–c; Supplementary Table 1). *E.coli* MMR proteins were purified and labeled with specific fluorophores similar to previous studies with minor modifications (Supplementary Fig. 2d; Supplementary Tables 1, 2)[13,14]. Injection of Cy3-MutL into the flow cell resulted in numerous bound particles that randomly diffused along the DNA (Supplementary Fig. 2e). The initial association of MutL with DNA appeared to be random along the entire length of mismatched DNA (Supplementary Fig. 2f), suggesting that the interaction is independent of the mismatch and DNA sequence.

The frequency of MutL-DNA interactions rapidly decreased as ionic strength was increased, with few if any interactions above ~80 mM total ionic strength (60 mM NaCl; Fig. 1b, left; Fig. 1c, black dots). These observations are consistent with previous work and suggest that any interaction between MutL and DNA is either non-existent or significantly shorter than the imaging frame-rate (100 ms) at physiological ionic strength[13,25]. Interestingly, once on the DNA the lifetime of MutL remained constant, and the diffusion coefficient ($D$) did not change significantly over a range of low ionic strength conditions (Fig. 1d, e; Supplementary Table 3). These results suggest that: (1) ion shielding of the DNA with increasing ionic strength results in a decreased $k_{on}$ (increased $K_D$) and a rapid decrease in the frequency of MutL-DNA interactions, (2) once bound MutL maintains continuous contact with the backbone consistent with rotation coupled diffusion during its movement along the DNA, and (3) the majority of MutL binding event do not result in a long-lived sliding clamp[13,15,51]. Together, these observations seem to imply a transient non-specific electrostatic interaction between MutL and the DNA backbone that is undetectable at physiological ionic strength.

To further establish whether the MutL PCC is responsible for these low ionic strength DNA interactions, we changed the single previously studied conserved Arg residue R266 to Glu [referred to as MutL(R266E)] as well as three simultaneous Arg residues (R162, R266 and R316) to Glu [referred to as MutL(R-E)] (Fig. 1a, green arrowheads, and black boxes; Supplementary Tables 1, 2)[24,49]. Genetic studies confirmed historical conclusions that expression plasmids containing the MutL(R266E) and MutL(R-E) mutations are unable to complement an *E. coli* Δ*mutL* mutant strain resulting in an elevated frequency of spontaneous Rif[r] mutations (Supplementary Fig. 3)[23,48]. Single-particle imaging by smTIRF showed that the MutL(R266E) and MutL(R-E) proteins displayed consistently fewer DNA interactions than wild-type MutL over a range of ionic strength (Fig. 1b, c). As might be expected, the MutL(R266E) protein containing a single altered conserved PCC residue appeared progressively less defective than MutL(R-E) protein

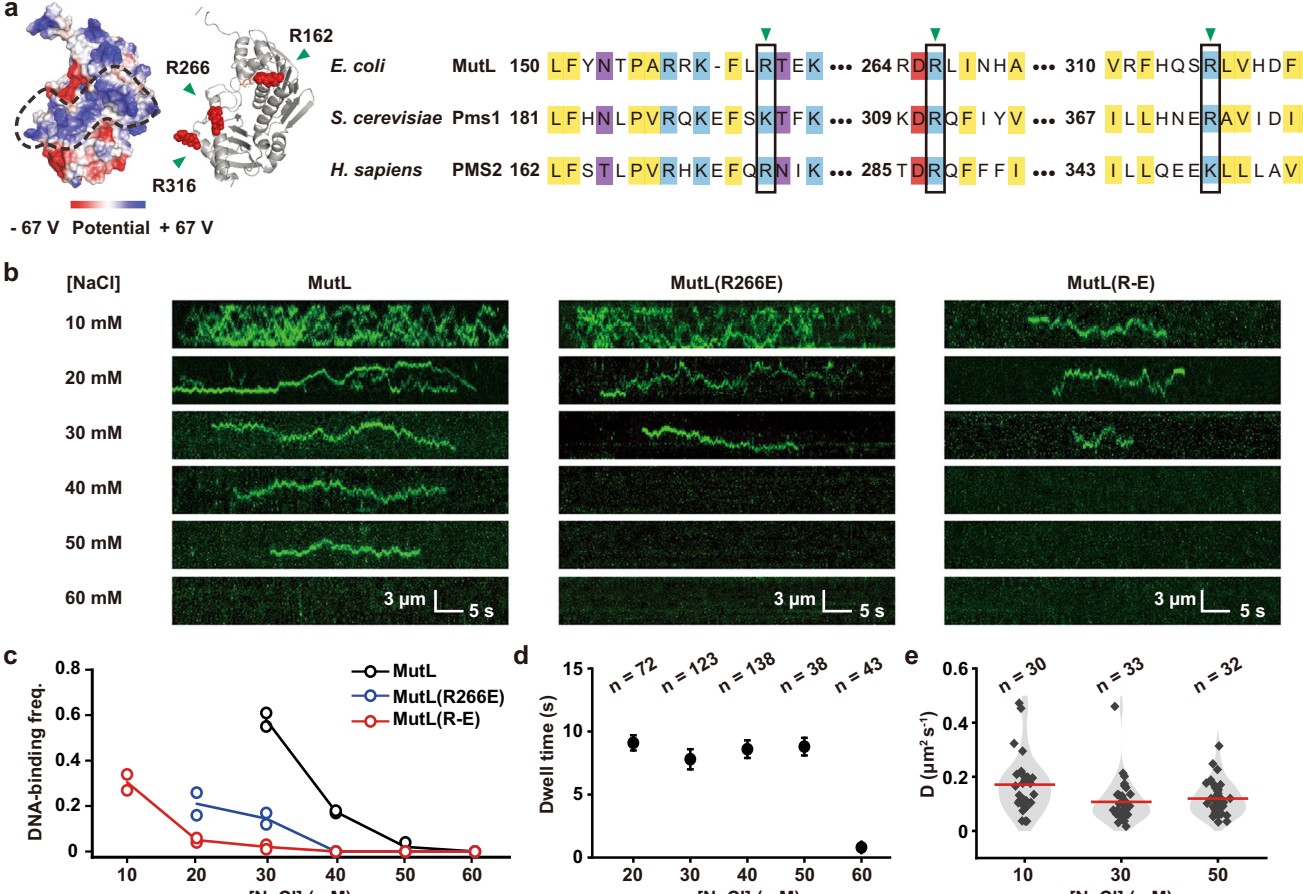

**Fig. 1 | The MutL NTD is responsible for low ionic strength DNA binding activity. a** Left: color-coded electrostatic surface potential diagram of MutL NTD identifying the positively charged cleft (PCC, black dashed circle; PDB ID: 1B62) with the location of previously examined DNA interaction residues indicated by green arrowheads. Right: sequence alignment of NTD PCC between *E.coli* MutL and the MutL homologs Pms1 and PMS2 of *S. cerevisae* and human, respectively. Previously examined DNA interaction residues indicated by green arrowheads and black rectangles. Conserved hydrophobic residues are shaded in yellow, basic in blue, acidic in red, and others in purple. **b** Representative kymographs showing the binding and diffusions of MutL, MutL(R266E) or MutL(R-E) (2 nM) on mismatched DNA under various ionic conditions. **c** The frequency of MutL, MutL(R266E) or MutL(R-E) (2 nM) bound to mismatched DNA under various ionic conditions. Open circles represent individual frequency from independent experiments and the mean frequencies are connected by lines. (For MutL binding at 30–60 mM NaCl: $n = 318, 319, 271, 249$, respectively; For MutL(R266E) binding at 20–60 mM NaCl: $n = 397, 394, 394, 394, 394$, respectively; For MutL(R-E) binding at 10–60 mM NaCl: $n = 315, 322, 188, 195, 246, 270$, respectively; $n =$ total number of DNA molecules examined). **d** The dwell time of MutL bound to DNA (mean ± s.e.) under various ionic conditions (n = number of events examined). **e** Violin plots of the diffusion coefficient (*D*) for MutL on mismatched DNA under various ionic conditions (black diamonds indicate individual molecules; red line indicates mean; *n* = number of events examined).

containing three altered conserved PCC residues. For example, at ~50 mM total ionic strength (30 mM NaCl) the MutL(R266E) protein displayed ~3-fold fewer DNA interaction events, while the MutL(R-E) protein displays 30-fold fewer DNA interaction events compared with wild-type MutL (Fig. 1b, c). No DNA interactions were observed by either of these proteins above ~60 mM total ionic strength (40 mM NaCl), which is well below physiological ionic strength (Fig. 1b, middle and right; Fig. 1c, blue and red dots). These results unambiguously connect the MutL PCC to the very low ionic strength DNA binding events similar to the previous reports[23,24,37,45,48]. However, the mechanical role of the MutL PCC in MMR seems uncertain since MutL DNA binding activity is completely absent at physiological ionic strength.

## DNA is essential for a MutS-MutL interaction

The increased mutation frequency found in cells containing MutL(R266E) or MutL(R-E) could result from the disruption of one or more MutL activities during MMR. These might include its association with MutS, MutH or UvrD as well as its ability to bind ATP and form a sliding clamp. We first examined the ATP binding and hydrolysis

(ATPase) and observed a small (1.5x) decrease in activity between MutL(R-E) ($0.19 \pm 0.01 \, min^{-1}$) and wild-type MutL ($0.29 \pm 0.01 \, min^{-1}$; Supplementary Fig. 4a). These results suggest that triple PCC Arg→Glu mutations do not substantially compromise ATP binding and hydrolysis. We then visualized MutS and MutL on the 18.4 kb mismatched DNA by smTIRF[13,14]. Injection of Cy5-MutS with ATP resulted in numerous long-live MutS particles that largely originated at the mismatch and randomly diffused along the DNA (Fig. 2a). These observations mimic previous results showing that mismatch recognition triggers the formation of dynamic ATP-bound MutS sliding clamps on DNA[15,17–21,52]. Co-injection of Cy3-MutL with Cy5-MutS resulted in frequent co-localization consistent with previous results that detailed the formation of an initial MutS-MutL complex on a mismatched DNA (Fig. 2b, left; Fig. 2c)[13,14,17]. However, substitution wild-type MutL with MutL(R266E) or MutL(R-E) at physiological ionic strength eliminated these initial MutS-MutL complexes, suggesting that MutL(R266E) and MutL(R-E) do not appropriately interact with MutS sliding clamps on the DNA (Fig. 2b, right; Fig. 2c).

There are three components associated with the formation of the initial MutS-MutL complexes: MutS sliding clamps, MutL, and DNA. To

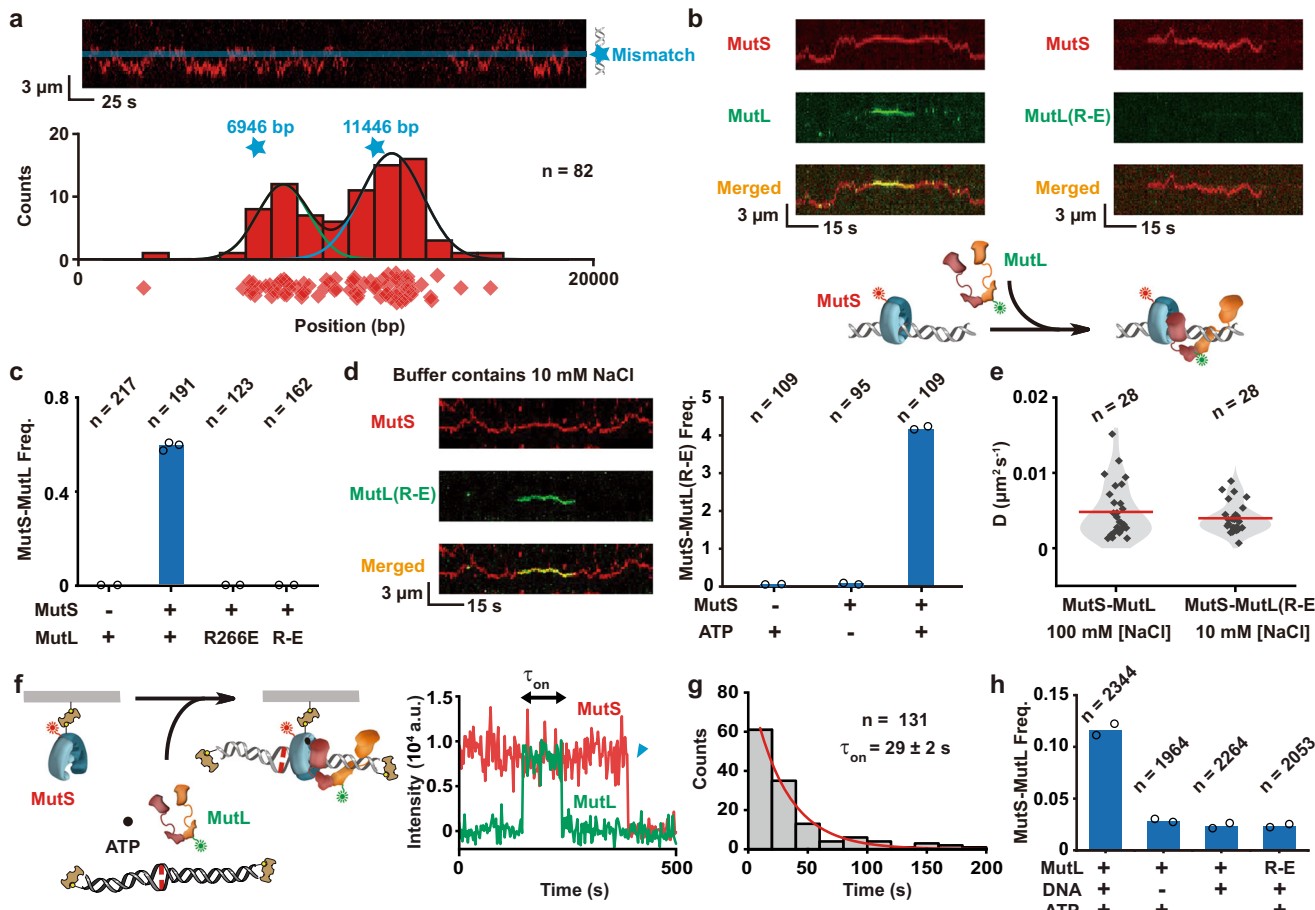

**Fig. 2 | MutS sliding clamp enhances loading of MutL onto DNA. a** Top: Representative kymograph showing two individual MutS sliding clamp events on a single mismatched DNA. Blue star and line indicate the position of the mismatch. Bottom: The distribution of individual initial binding location for MutS bound to the DNA. Diamonds represent individual starting point and the blue stars indicate the two possible positions of the mismatch on randomly attached substrate DNAs (*n* = number of events examined). Gaussian fits to the distribution of individual initial binding location are shown as lines. **b** Representative kymographs (top) and illustration (bottom) showing that a Cy5-MutS sliding clamp can recruit wild-type Cy3-MutL (left) onto the mismatched DNA, while there is no recruitment of MutS-MutL(R-E) (right) at physiological ionic conditions (buffer containing 100 mM NaCl, ~120 mM). **c** The frequency of MutS-MutL complexes under physiological ionic conditions (~120 mM, *n* = total number of DNA molecules examined). **d** Representative kymographs (left) and frequency of MutS-MutL(R-E) complexes

(right) under low ionic strength conditions (buffer containing 10 mM NaCl, ~30 mM, *n* = total number of DNA molecules examined). **e** Violin plots of the diffusion coefficient (*D*) for the MutS-MutL (~120 mM) and the MutS-MutL(R-E) complex (~30 mM) (red line = mean; *n* = number of events examined). **f** An illustration (left) and a representative trajectory (right) showing a Cy5-MutS-biotin bound to a flow cell surface via neutravidin with a Cy3-MutL co-localization event in the presence of mismatched DNA and ATP. Blue triangle indicates Cy5 photobleaching. **g** Distribution of dwell times for co-localized MutS-MutL complexes in the presence of mismatched DNA and ATP. Data were fit to a single exponential decay to derive the average lifetime ($\tau_{on}$, mean ± s.e.; *n* = number of events examined). **h** The frequency of MutS-MutL co-localization with various component additions shown below (*n* = total number of Cy5-bio-MutS molecules examined). All panels: open circles represent individual frequency from independent experiments and columns represent mean values from at least two experiments.

probe the role of a MutL-DNA interaction, we reduced the ionic strength (10 mM NaCl) to restore partial binding between MutL(R-E) and DNA (Fig. 1c). Under these conditions, numerous MutS- and ATP-dependent MutS-MutL(R-E) complexes were detected on mismatched DNA (Fig. 2d). Importantly, the diffusion coefficient of the MutS-MutL(R-E) complex ($D_{MutL(R-E)} = 0.004 \pm 0.002 \, \mu m^2 \, s^{-1}$) appeared to be identical to the MutS-MutL complex ($D_{MutL} = 0.005 \pm 0.004 \, \mu m^2 \, s^{-1}$; Fig. 2e; Supplementary Table 3)[13]. These observations are consistent with the conclusion that an interaction between the MutL PCC and DNA is necessary for the formation of the initial MutS-MutL complex.

To further probe the biophysical requirements for the formation of the initial MutS-MutL interaction, we developed a quantitative single-molecule surface-bound protein interaction system. MutS containing a C-terminal biotin and Cy5 fluorophore was purified and shown to efficiently form typical sliding clamps on 18.4 kb mismatched DNA by smTIRF (Supplementary Fig. 4b). Immobilization of bio-Cy5-MutS on a passivated smTIRF surface via biotin-neutravidin resulted in

numerous single molecules that could be easily visualized in the Cy5 channel (Supplementary Fig. 4c). Injection of Cy3-MutL with ATP and a biotin-neutravidin blocked-end 59-bp mismatched DNA resulted in co-localization of Cy5-MutS with Cy3-MutL (Fig. 2f) that displayed a lifetime ($\tau_{on}$) of $29 \pm 2$ s (Fig. 2g), which is similar to previous smTIRF studies that detailed the formation of initial MutS-MutL complexes on an 18 kb mismatched DNA ($\tau_{on} = 32 \pm 2$ s)[13]. The MutS-MutL interaction required both ATP and DNA and was eliminated when MutL(R-E) was substituted for wild-type MutL (Fig. 2h). These studies imply that the formation of an initial MutS-MutL complex requires simultaneously interactions between MutS, DNA, and a functional MutL PCC. The simplest interpretation of these results suggests that the surface-immobilized bio-Cy5-MutS must first capture the mismatched DNA, which in the presence of ATP forms a sliding clamp that retains the mismatched DNA by virtue of its blocked ends[15,17–21]. The MutS sliding clamp-DNA complex may then recruit MutL forming a MutS-MutL complex. Because the available structures indicate that the MutS-MutL

interaction region is located at a significant distance from MutL PCC[22,23], we conclude that dynamic MutS-MutL complex formation in our smTIRF system entails separate contacts between MutL with an operational MutS sliding clamp and MutL with the DNA.

## The MutL PCC is exploited to form an ATP-bound MutL sliding clamp

Consistent with our previous observations, an interaction between MutS with MutL on a mismatched DNA is necessary to produce long-lived fast-diffusing ATP-bound MutL sliding clamps on an 18.4 kb mismatched DNA (Fig. 3a, left; Fig. 3b)[13,14]. However, fast-diffusing MutL(R-E) particles were not observed under similar physiological ionic conditions (Fig. 3a, right; Fig. 3b). To explore the function of the MutL PCC, we examined ionic conditions and MMR component requirements that resulted in fast-diffusing MutL sliding clamps. For these studies, we first included MMR components under variable ionic strength conditions, then switched to physiological ionic strength by a buffer exchange within the single-molecule flow cell and recorded the frequency and properties of fast diffusing MutL sliding clamps (Fig. 3c). As expected, the highest frequency of fast-diffusing MutL sliding clamps across all initial ionic conditions was when wild-type MutS and MutL were present with ATP (0.567–0.073 between 10 and 100 mM NaCl, respectively, or 57–7% of the DNA molecules with 10 nM MutL; Fig. 3d; Supplementary Table 4)[13]. Substitution of wild-type MutL for the triple Arg→Glu mutant protein MutL(R-E) only resulted in fast diffusing sliding clamps at very low ionic strength (≤30 mM NaCl), while the single mutant MutL(R266E) resulted in fast diffusing sliding clamps up to an initial concentration of 50 mM NaCl (Fig. 3d; Supplementary Table 4). These results mimicked the pattern of ionic strength-dependent recovery of DNA interaction activity by these mutant MutL proteins (compare Fig. 3d with Fig. 1c; Fig. 3e). While MutS and ATP are generally required to load MutL sliding clamps, we note that MutL alone forms fast-diffusing sliding clamps on a significant number of DNA molecules in very low ionic strength (0.25 at 10 mM NaCl, or 25% of the DNA molecules with 10 nM MutL; Fig. 3d; Supplementary Table 4). Together, these results suggest MutL possesses an intrinsic low-probability capacity to form ATP-bound fast diffusing sliding clamps that is substantially accelerated by MutS sliding clamps on the DNA.

The MutL(R-E) sliding clamps appeared to diffuse somewhat faster along the DNA than wild-type MutL sliding clamps (Fig. 4a). These results are consistent with the hypothesis that the wild-type MutL PCC may undergo incidental short-lived interactions with the DNA backbone, modestly slowing diffusion. Such transient interactions are likely moderated in the MutL(R-E) PCC mutant protein at physiological ionic strength accounting for an increased diffusion coefficient. In addition, the MutL(R-E) sliding clamps appear slightly less stable on the DNA than wild-type MutL sliding clamps, although this effect could be a consequence of the buffer-switch conditions that might unduly influence MutL(R-E) sliding clamps (Fig. 4b). Once on the DNA together with MutS, the MutL sliding clamps regularly experience dynamic association-dissociation events that alter their diffusion properties (Fig. 4c)[13]. We found the association lifetime of the oscillating MutS-MutL sliding clamp complexes was similar to our previous report ($\tau_{MutS-MutL\ SC} = 20 \pm 2$ s; Fig. 4d, left)[13]. However, the association lifetime of the oscillating MutS-MutL(R-E) sliding clamp complex was significantly shorter ($\tau_{MutS-MutL(R-E)\ SC} = 0.7 \pm 0.03$ s; Fig. 4d, right). These observations are consistent with the conclusion that the wild-type MutS-MutL sliding clamp complex naturally engages the MutL PCC with the DNA backbone altering its lifetime and diffusion properties, while a similar engagement is largely refractory with the MutL(R-E) protein. Taken together we conclude the DNA-bound MutS sliding clamp creates a physical environment that promotes complex assembly with the MutL NTD and concurrent positioning of the PCC on the DNA backbone[22,23] that leads to both clamp loading and dynamic MutS-MutL association-dissociation.

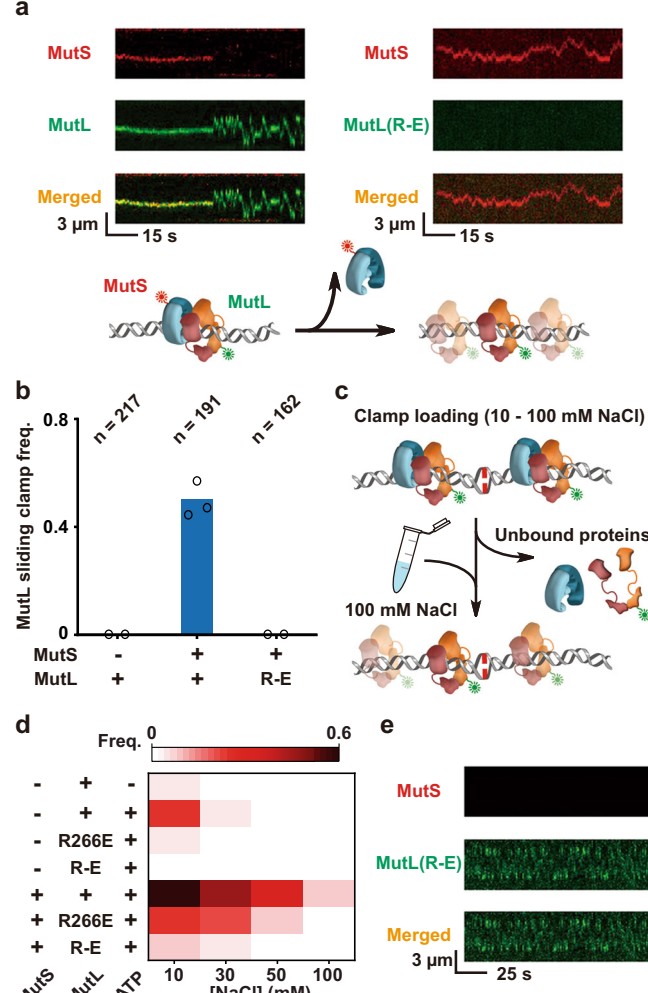

**Fig. 3 | MutS and DNA cooperate with the MutL NTD positively charged cleft to load a MutL sliding clamp onto the mismatched DNA. a** Representative kymographs (top) and illustration (bottom) showing the formation of a fast-diffusion Cy3-MutL sliding clamp from a Cy5-MutS/Cy3-MutL complex (left), while no complexes or Cy3-MutL(R-E) sliding clamps (right) were observed under physiological ionic conditions (-120 mM). Cy5-MutS and Cy3-MutL color-coded as described in Fig. 2b. **b** The frequency and component requirements for fast-diffusing MutL sliding clamps on the mismatched DNA under physiological ionic condition (-120 mM, $n$ = total number of DNA molecules examined). **c** An illustration showing the reaction and wash sequence to examine the frequency of fast-diffusing Cy3-MutL clamps loaded onto a mismatched DNA under various ionic conditions. MutS and Cy3-MutL color-coded as described in Fig. 2b. **d** Heat map of the frequency of fast-diffusing MutL sliding clamps with various included MMR components (left) and ionic loading conditions (bottom; see: Supplementary Table 4). **e** Representative kymographs showing MutS with DNA can load a fast-diffusing MutL(R-E) sliding clamp under very low ionic strength conditions (-30 mM).

## ATP hydrolysis releases MutL sliding clamps from the mismatched DNA

Once formed, MutL sliding clamps appear to randomly move along the DNA[13,14,25]. It is formally possible that this motion could involve cycles of ATP binding and hydrolysis. To examine this prospect, we developed a two-step clamp loading procedure, where the MutS sliding clamps were loaded first in the presence of ATP and then unbound proteins, as well as ATP, were removed by a buffer exchange (Fig. 5a). MutL was then introduced in the presence of ATP or the non-hydrolyzable ATP-analog adenylyl-imidodiphosphate (AMP-PNP; Fig. 5a). This strategy resulted in an equal frequency of mismatched DNAs containing fast-diffusing MutL sliding clamps compared to the

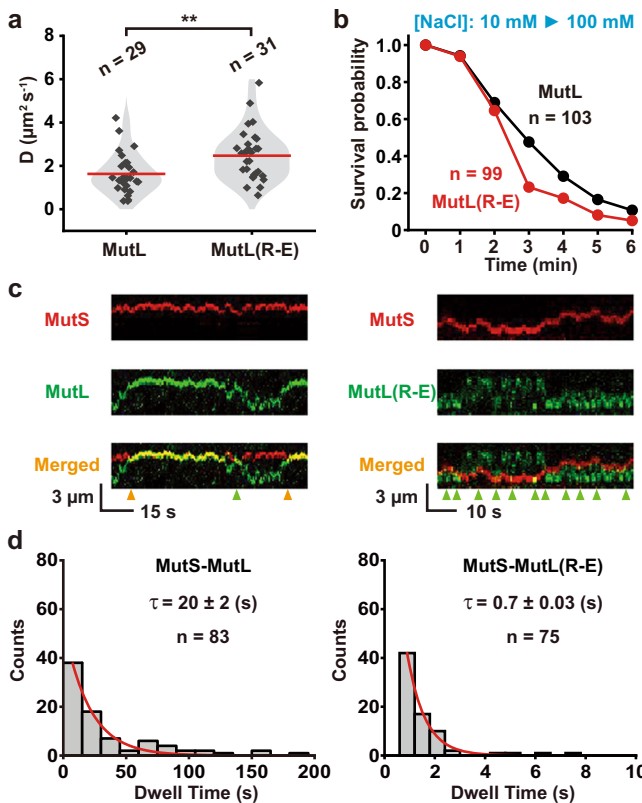

**Fig. 4 | The biophysical properties of fast-diffusing wild-type MutL and MutL(R-E) sliding clamps on mismatched DNA. a** Violin plots of the diffusion coefficient (*D*) for wild-type MutL and MutL(R-E) sliding clamps examined after buffer exchange to physiological ionic strength (~120 mM, *n* = number of events examined). Statistical significance and *p* value (*p* = 0.003) were determined by two-sided *t*-test for the comparison between two groups. '**' represents 0.001 < *p* < 0.01. **b** Survival probability of wild-type MutL or MutL(R-E) sliding clamps determined after buffer exchange to physiological ionic strength (~120 mM, *n* = total number of molecules). **c** Representative kymographs showing association-dissociation of MutS and MutL sliding clamps to form MutS-MutL complex (left). Note only transient interactions between MutS and MutL(R-E) sliding clamps (right). Orange arrowheads indicate the MutS-MutL collisions with stable interactions. Green arrowheads indicate the MutS-MutL collisions with transient interactions. **d** The distribution of dwell times for MutS-MutL or MutS-MutL(R-E) complexes shown in **c**. Data were fit to a single exponential decay to derive the average lifetime (τ, mean ± s.e.; *n* = number of events examined).

inclusion of ATP (Fig. 5b). Importantly, the MutL sliding clamps formed with ATP or AMP-PNP diffused along the entire length of the mismatch DNA (Fig. 5b). However, the AMP-PNP-bound MutL sliding clamps displayed a significantly longer lifetime on the mismatched DNA ($t_{1/2 \cdot AMP-PNP}$ = 51.9 min, compared to $t_{1/2 \cdot ATP}$ = 8.3 min, Fig. 5c). We attribute the difference in MutL sliding clamp lifetime compared to our previous work[13] to the additional time required for wash cycles prior to starting the lifetime clock. Taken together these observations are consistent with the conclusion that the movement of MutL sliding clamps on the mismatched DNA is mediated by random thermal diffusion and that ATP hydrolysis leads to dissociation of the N-terminal domains resulting in release the MutL sliding clamps from the DNA (Supplementary Fig. 1b).

**The MutL PCC is not required to bind MutH but is employed by the MutH hemimethylated GATC endonuclease**

Previous studies demonstrated an interaction between the MutL sliding clamp with MutH on the mismatched DNA, which together with MutS was proposed to perform the search and strand incision of hemimethylated GATC sites[13,14]. To examine the role of the MutL PCC

on this downstream MMR activity, we incorporated MutH into the smTIRF reactions initiation by MutS and MutL (Fig. 6). No complexes with MutH were observed at physiological ionic strength when MutL(R-E) was substituted for wild-type MutH (Fig. 6a; Supplementary Table 5). These results suggest that the MutL(R-E) protein is unable to mediate communication between mismatch recognition and MutH hemimethylated GATC strand incision.

We reasoned that the downstream interaction defect between MutL(R-E) and MutH was likely due to the inability of the MutS clamp loader to assemble MutL(R-E) stable sliding clamps on the mismatched DNA at physiological ionic strength (Fig. 3). To test this hypothesis, we initially incubated MutS and Cy3-labeled MutL(R-E) at very low ionic strength (10 mM NaCl) in the smTIRF system. Under these conditions stable ATP-bound MutL(R-E) sliding clamps can be loaded by the MutS sliding clamp onto the mismatched DNA (Figs. 3, 6b). The buffer was then exchanged to physiological ionic strength (~120 mM; 100 mM NaCl) and Cy5-labeled MutH was injected into the flow cell (Fig. 6b). Numerous MutL(R-E)-MutH complexes were observed that depended on the addition of MutS (Figs. 3d, 6c; Supplementary Table 5), strongly supporting the conclusion that mutation of the MutL PCC does not influence stable interactions between a MutL sliding clamp with MutH[13,14].

To examine the role of the MutL PCC in MutH hemimethylated GATC incision activity, a 4.8 kb circular plasmid DNA was constructed that contained a single G/T mismatch with an adjacent hemi-methylated GATC site marked by a nearby Cy3-fluorophore as well as a distant biotin residue (Cy3-biotin mismatched DNA; Supplementary Fig. 5a). The G/T mismatch included overlapping NsiI and NcoI restriction sites, where NcoI is the native duplex DNA restriction site. Diagnostic restriction digests demonstrate that the presence of a mismatch results in resistance to both NsiI and NcoI (Supplementary Fig. 5b)[53,54]. Moreover, we demonstrated selective precipitation of the mismatched plasmid containing the biotin residue when mixed with streptavidin-coated superparamagnetic beads (SA beads; Supplementary Fig. 5c). Incubation of the Cy3-biotin mismatched plasmid DNA with MMR proteins followed by digestion with BbvCI and denaturing PAGE analysis indicates that the normal 72 nt fragment is most efficiently incised at the adjacent hemimethylated GATC site when MutS, MutL and MutH are present together, producing a diagnostic 58 nt Cy3-labeled product (Supplementary Fig. 5d). We noted that MutL with MutH may also generate hemimethylated GATC incision products in the absence of MutS (Supplementary Fig. 5d). As expected, the MutS and MutL proteins were substantially enriched with the Cy3-biotin mismatched DNA precipitated by SA beads (Supplementary Fig. 5e). These results demonstrate that MutH hemimethylated GATC incision as well as the correlated presence of stable MutS and MutL sliding clamps can be detected using the Cy3-biotin mismatched DNA substrate.

The contribution of the MutL PCC to MutH hemimethylated GATC incision activity was examined utilizing a stepwise MMR protein loading protocol, where MutS was first incubated with the Cy3-biotin mismatched DNA (Fig. 6d). Unbound MutS was then removed by precipitation of the Cy3-biotin mismatched DNA with SA beads, and MutL was added with AMP-PNP to load long-lived MutL sliding clamps (Figs. 5c, 6d). The Cy3-biotin mismatched DNA constituents were exchanged again by SA bead precipitation to simultaneously remove free MutL protein and introduce the MutH protein in ~120 mM ionic strength buffer (100 mM NaCl; Fig. 6d). Examination of the proteins bound to the SA bead precipitated Cy3-biotin mismatched DNA prior to the addition of MutS shows that MutS and MutL sliding clamps remain firmly associated with the circular plasmid DNA following the loading process (Fig. 6e, 100 mM NaCl)[13,14]. As expected MutS sliding clamps were required to load MutL sliding clamps onto the Cy3-biotin mismatched DNA similar to previous studies (Fig. 6e, 100 mM NaCl)[13,14]. The addition of MutH endonuclease resulted in a significant

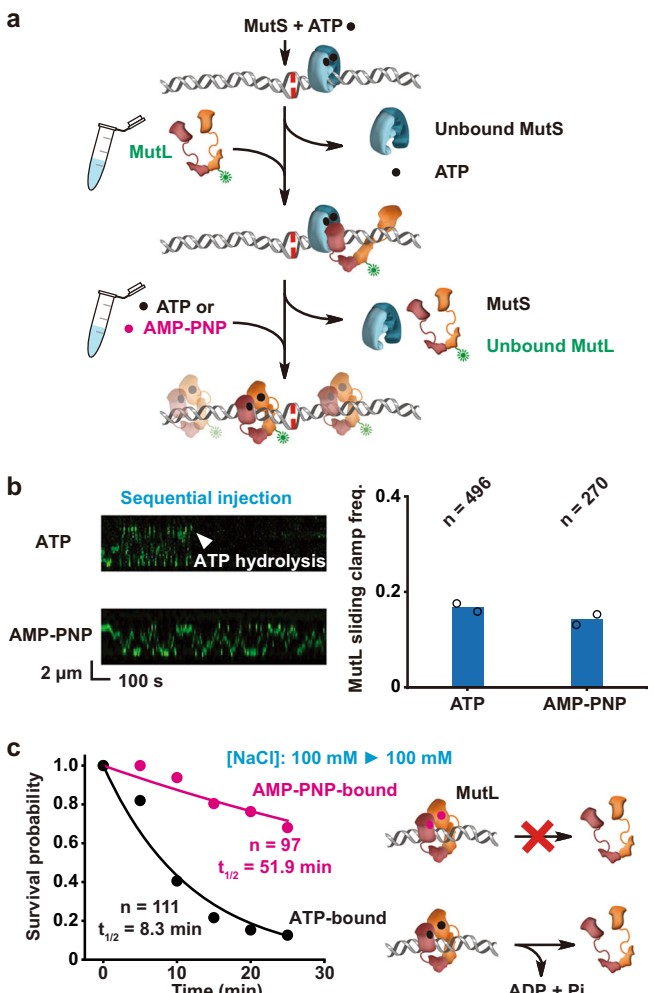

**Fig. 5 | ATP hydrolysis releases MutL sliding clamps from the mismatched DNA.**
**a** An illustration of the two-step injection process that loads ATP-bound MutS sliding clamps and then ATP- or AMP-PNP-bound MutL sliding clamps onto the mismatched DNA. MutS and Cy3-MutL color-coded as described in Fig. 2b.
**b** Representative kymographs (left) and frequency (right) of MutL sliding clamps on the mismatched DNA examined by two-step injection ($n$ = total number of DNA molecules examined). **c** Left: Survival probability of ATP-bound or AMP-PNP-bound MutL sliding clamp. Data were fit by exponential decay functions to obtain a survival half-life ($t_{1/2}$, mean ± s.e; $n$ = number of events examined); Right: Illustrations showing that MutL color-coded as described in Fig. 2b bound by AMP-PNP (magenta dots, top) is substantially resistant to hydrolysis and ring opening, while ATP (black dots, bottom) hydrolysis opens the MutL ring-like clamp to dissociate the protein from DNA.

fraction of incised hemimethylated GATC product when MutS, MutL and MutH were present (Fig. 6f, 100 mM NaCl; Supplementary Fig. 5f, top). These observations are consistent with historical studies and demonstrate that mismatch-dependent strand-specific incision at hemimethylated GATC sites requires MutS, MutL, and MutH at physiological ionic strength[13,14,55,56].

To examine the influence of the MutL PCC, we exploited the ability of MutS sliding clamps to load the mutant MutL(R-E) protein at very low ionic strength (Fig. 3d). We first confirmed that MutS and MutL(R-E) sliding clamps were specifically and stably loaded onto the Cy3-biotin mismatched DNA by precipitation with SA beads (Fig. 6e, 10 mM NaCl). However, the addition of MutH protein at physiologically significant ionic strength (-120 mM; 100 mM NaCl) did not result in any detectable hemimethylated GATC incision products (Fig. 6f, 10 mM NaCl, R-E lanes). These results suggest that the MutL PCC is exploited

by the MutH endonuclease during the strand-specific incision processes. Interestingly, when wild-type MutL sliding clamps were loaded at very low salt onto the Cy3-biotin mismatched DNA in the absence of MutS (10 mM NaCl; Fig. 6e), we observed significant MutH hemimethylated GATC endonuclease activity that appeared equivalent to the endonuclease activity observed at physiological ionic strength in the presence of MutS plus MutL (Fig. 6f; Supplementary Fig. 5f; see Discussion below).

### The MutL PCC is utilized during UvrD helicase capture and DNA unwinding

We confirmed our previous work that demonstrated MutL sliding clamps capture the UvrD helicase at a strand scission, enhancing both the lifetime and unwinding of mismatched DNA by smTIRF (Fig. 7a)[13,14]. In contrast, we observed no colocalization of MutL(R-E) with UvrD or any evidence of DNA unwinding activity (Fig. 7a; Supplementary Table 5). These results are consistent with the inability of MutL(R-E) to be loaded onto the DNA as a stable sliding clamp at physiological ionic strength (Fig. 3d). When MutL(R-E) was loaded at very low ionic strength (10 mM NaCl) and interaction with UvrD was examined at physiologically significant ionic strength following a buffer exchange (-120 mM, 100 mM NaCl; Fig. 7b), only a small fraction of colocalization and unwinding was observed that was dependent on the presence of MutS (Fig. 7c; Supplementary Table 5). However, when MutL(R-E) sliding clamps were loaded onto DNA at very low ionic strength (10 mM) and unwinding activity was also examined at low ionic strength (50 mM NaCl) where MutL(R-E) DNA binding activity remains at least partially functional (Fig. 1c), we observed colocalization with UvrD that was significantly stimulated by including MutS (Supplementary Fig. 6a). These results suggest that the MutL PCC enhances UvrD helicase capture at the strand scission.

Our previous work clearly demonstrated that MutH-dependent strand scissions flanking a mismatch resulted in extremely efficient MMR excision that did not require a ssDNA exonuclease activity[14]. This observation solved a decades-old genetic conundrum of why mutation of all the consensus ssDNA exonucleases was more than 100-fold less adept at reducing MMR in vivo than mutation of any core MMR gene MutS, MutL, MutH, and UvrD[57]. To quantitatively examine UvrD helicase ssDNA displacement, we modified the Cy3-biotin mismatched DNA to retain 3′ and 5′ strand scissions flanking the 72 nt oligonucleotide containing the mismatch and Cy3 fluorophore (Supplementary Fig. 6b). Unwinding activity was observed as either the release of the 72 nt Cy3-mismatch-containing ssDNA fragment following SA bead precipitation of the Cy3-biotin mismatched DNA (Fig. 7d) or the loss of the Cy3-label from the circular plasmid DNA (Supplementary Fig. 6c). As shown previously, MutS, MutL and UvrD are required to displace the mismatch-containing strand between two strand scissions at physiologically significant ionic strength (-120 mM; Fig. 7e, 100 mM NaCl; Supplementary Fig. 6c)[14]. In contrast, under conditions in which MutS is capable of loading MutL(R-E) sliding clamps, no helicase strand displacement was observed (Figs. 3d, 7e, 10 mM NaCl). These results provide additional evidence that the MutL PCC is essential to capture and promote processive DNA unwinding by the UvrD helicase.

To test the requirement of MutS sliding clamps for UvrD helicase strand displacement, we loaded stable ATP-bound wild-type MutL sliding clamps alone onto the Cy3-biotin mismatched DNA at very low ionic strength (Fig. 7e, 10 mM NaCl). Buffer exchange to physiologically significant ionic strength (-120 mM; 100 mM NaCl) and addition of UvrD helicase resulted in robust strand displacement that appeared identical to a complete MutS, MutL, UvrD reaction (Fig. 7e; compare MutS, MutL, UvrD in 100 mM NaCl to MutL plus UvrD in 10 mM NaCl lane). These results are consistent with the conclusion that solitary MutL sliding clamps can promote UvrD unwinding and displacement of the mismatched DNA strand.

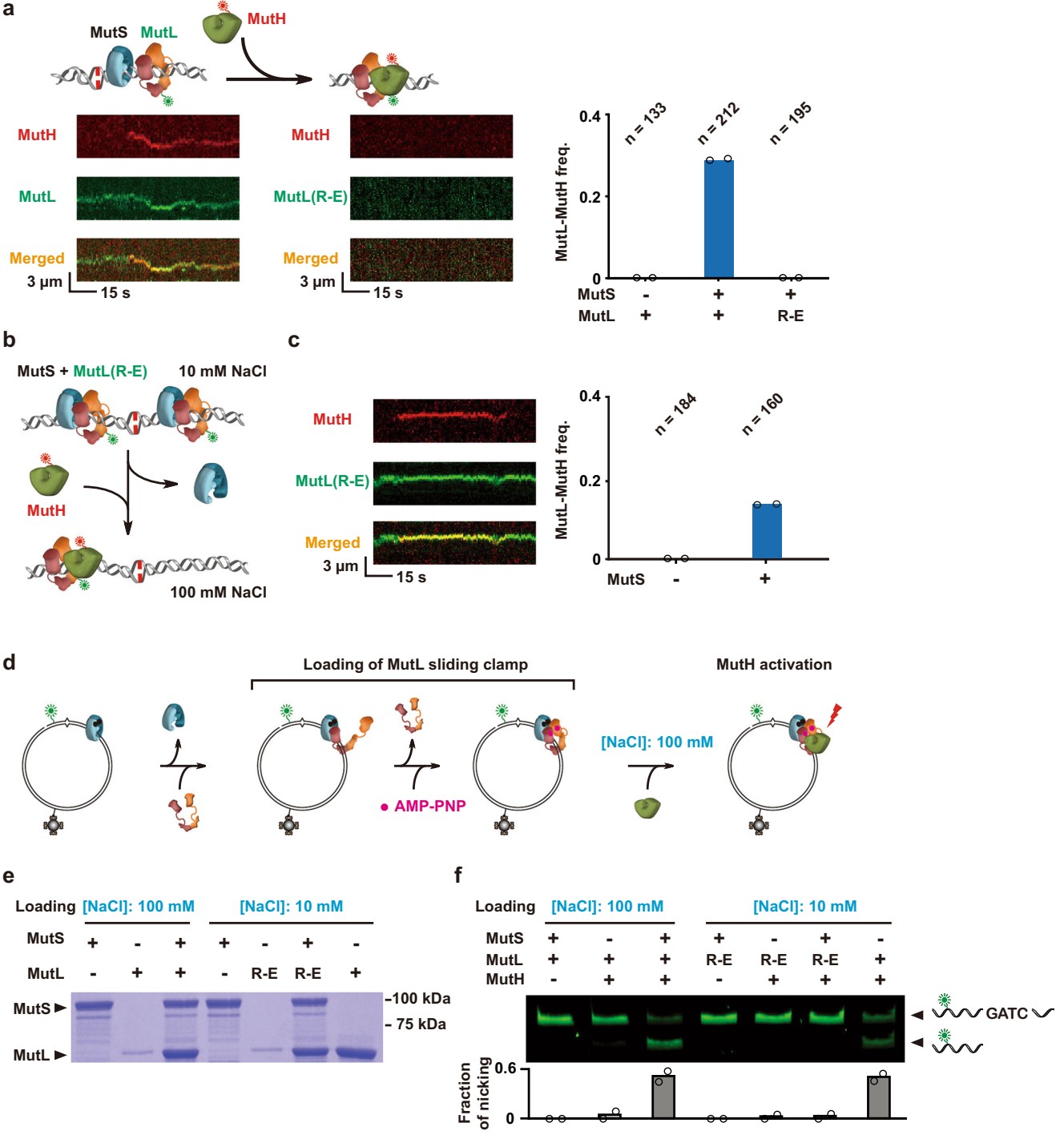

**Fig. 6 | The MutL NTD positively charged cleft does not affect MutL-MutH binding but does enhance the MutL-MutH complex search and incision of hemimethylated GATC sites. a** Left: Illustration and representative kymographs showing a Cy3-MutL sliding clamp color-coded as described in Fig. 2b interacting with a Cy5-MutH endonuclease (green), while no interaction between Cy3-MutL(R-E) and Cy5-MutH was observed under physiological ionic conditions (~120 mM). Right: The frequency of MutL-MutH complex formation under physiological ionic conditions (~120 mM; $n$ = total number of DNA molecules examined). **b** Illustration of the loading sequence for Cy3-MutL(R-E) sliding clamps onto the mismatched DNA followed by the addition of Cy5-MutH. MutS, Cy3-MutL(R-E) and Cy5-MutH color-coded as described in Figs. 2b, 6a. **c** Left: Representative kymograph of Cy3-MutL(R-E)/Cy5-MutH complex on a mismatched DNA; Right: the frequency of Cy3-MutL(R-E)/Cy5-MutH complexes in the absence/presence of MutS (n = total number of DNA molecules examined).

**d** Illustration of the loading sequence to examine proteins bound and MutH endonuclease activity utilizing the circular Cy3-biotin mismatched DNA substrate. MutS, MutL, MutH and AMP-PNP color-coded as described in Figs. 2b, 5c, 6a, with biotin shown in gray. **e** Coomassie-stained SDS-PAGE gel of MMR proteins precipitation with the circular Cy3-biotin mismatched DNA substrate. Included MMR components and NaCl loading concentration shown above. The MutS and MutL location is marked on the left. Background protein bands arise from minor MutS or MutL purification contaminants or proteolytic products during the SA-bead precipitation process. **f** Denaturing PAGE gel (top) and densitometry quantification from two independent experiments (bottom) that examined the MutH hemimethylated GATC endonuclease activity on the circular Cy3-biotin mismatched DNA substrate. Included MMR components and NaCl loading concentration shown above. The location of the intact and incised mismatch-containing fragment is shown on the right.

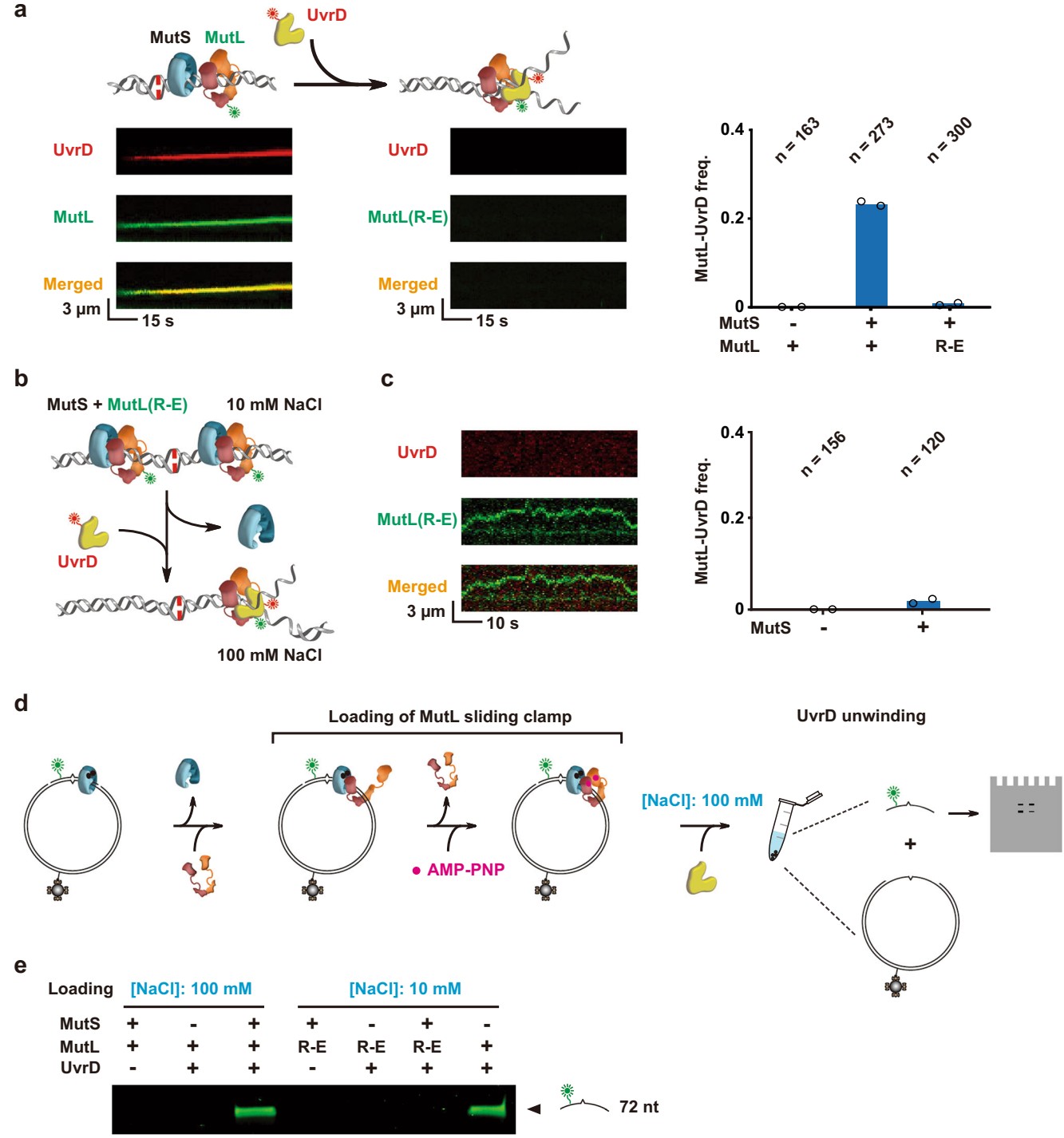

**Fig. 7 | The MutL NTD positively charged cleft enhances capture of the UvrD helicase. a** Left: Illustration (top) and representative kymographs (bottom) showing a Cy3-MutL sliding clamp loaded by an unlabeled MutS sliding clamp (color-coded as described in Fig. 2b) interacting with a Cy5-UvrD helicase (yellow), while no interaction was detected between Cy3-MutL(R-E) and Cy5-UvrD under physiological ionic conditions (-120 mM). Right: The frequency of Cy3-MutL/Cy5-UvrD complex formation at physiological ionic conditions (-120 mM; *n* = total number of DNA molecules examined). **b** An illustration showing the loading of Cy3-MutL(R-E) by MutS at very low ionic strength (-30 mM), followed by buffer exchange to observe the interaction between a Cy3-MutL(R-E) sliding clamp with Cy5-UvrD. MutS, MutL and UvrD are color-

coded as shown in Fig. 2b and above. **c** Representative kymograph (left) and frequency of Cy3-MutL(R-E)/Cy5-UvrD complexes (right) examined by the buffer exchange experiment shown in **b** (*n* = total number of DNA molecules examined). **d** Illustration of the loading sequence to examine UvrD helicase activity utilizing the circular Cy3-biotin mismatched DNA substrate. MutS, MutL, UvrD and AMP-PNP color-coded as described in Figs. 2b, 5c and above, with biotin shown in gray. **e** Denaturing PAGE gel of the UvrD helicase unwinding of the 72 nt Cy3-mismatch-containing ssDNA fragment following SA bead precipitation of the Cy3-biotin mismatched DNA. Included MMR components and NaCl loading concentration shown above. The location of the 72 nt mismatch-containing fragment is shown on the right.

## Discussion

It has been known for decades that MLH/PMS proteins function as mediators that connect MSH mismatch recognition to the strand excision processes which are essential for accurate MMR[1–4,58]. However, the detailed progression of these mediator functions has been a significant puzzle. Much of this uncertainty can be traced to the absence of a complete MLH/PMS structure, which to date has only revealed the N-terminal GHKL ATPase and C-terminal dimerization domain[24,35–39]. A large linker peptide connecting the N- and C-terminus of MLH/PMS proteins appears to be intrinsically disordered and refractory to current structural analysis techniques (Supplementary Fig. 1a). As a consequence, several MMR models have proposed, among other things, ATP-dependent folding of the intrinsically disordered linker (IDL)[40] and/or the emergence of extensive DNA binding activities as part of the MLH/PMS mediator function[23,24,37,44–48]. Such activities were projected to create a static MSH-MLH/PMS complex at or near the mismatch to accomplish MMR[3,41–43].

Previous work from our group demonstrated that mismatch recognition by *E.coli* MutS and the human homolog MSH2-MSH6 results in the formation of a stable (~3 min) sliding clamp on the mismatched DNA (Fig. 8a, left)[13,15,17,19,20,25]. Single-molecule analysis showed that the MSH sliding clamps subsequently recruit and ultimately load *E.coli* MutL and human MLH1-PMS2 onto the mismatched DNA as a cascade of sliding clamps (Fig. 8a, center and right)[13,25]. These real-time

studies recorded the MSH and MLH/PMS sliding clamps in continuous dynamic motion on the DNA[13–15,20,21,25,52,59], with no evidence of IDL ordering that should in theory constrict the MLH/PMS donut hole size and significantly slow its diffusion along the DNA[40]. Indeed, the genetic evidence appears consistent with a mostly disordered IDL for much of the MMR processes[13,14,25,27,60]. Intriguingly, the large donut hole and long lifetime of human MLH1-PMS2 suggested it should be capable of transiting most proteins that might bind to the DNA, including assembled or partially assembled nucleosomes; permitting the completion of downstream MMR processes even in the presence of chromatin factors[61,62].

Nevertheless, historical studies have demonstrated that MLH/PMS proteins including the *E.coli* MutL protein were capable of binding DNA, at least under the very low ionic strength conditions where it was studied[23,24,37,45,48]. Moreover, mutations of conserved N-terminal MutL PCC Arg/Lys residues effectively eliminated this DNA binding activity resulting in elevated mutation rates in vivo that were typical of MMR defects[23,37,48,49]. Companion work also showed MutL PCC-dependent activation of the *E.coli* MutH hemimethylated GATC endonuclease and UvrD helicase activities at low ionic strength in vitro, clearly implicating this DNA binding region in MMR processes[37,49,63]. However, these decades-old studies did not probe the interactive progressions of MutS, MutL, MutH or UvrD with DNA at physiological ionic strength nor did these studies anticipate the formation of a MutL sliding clamp

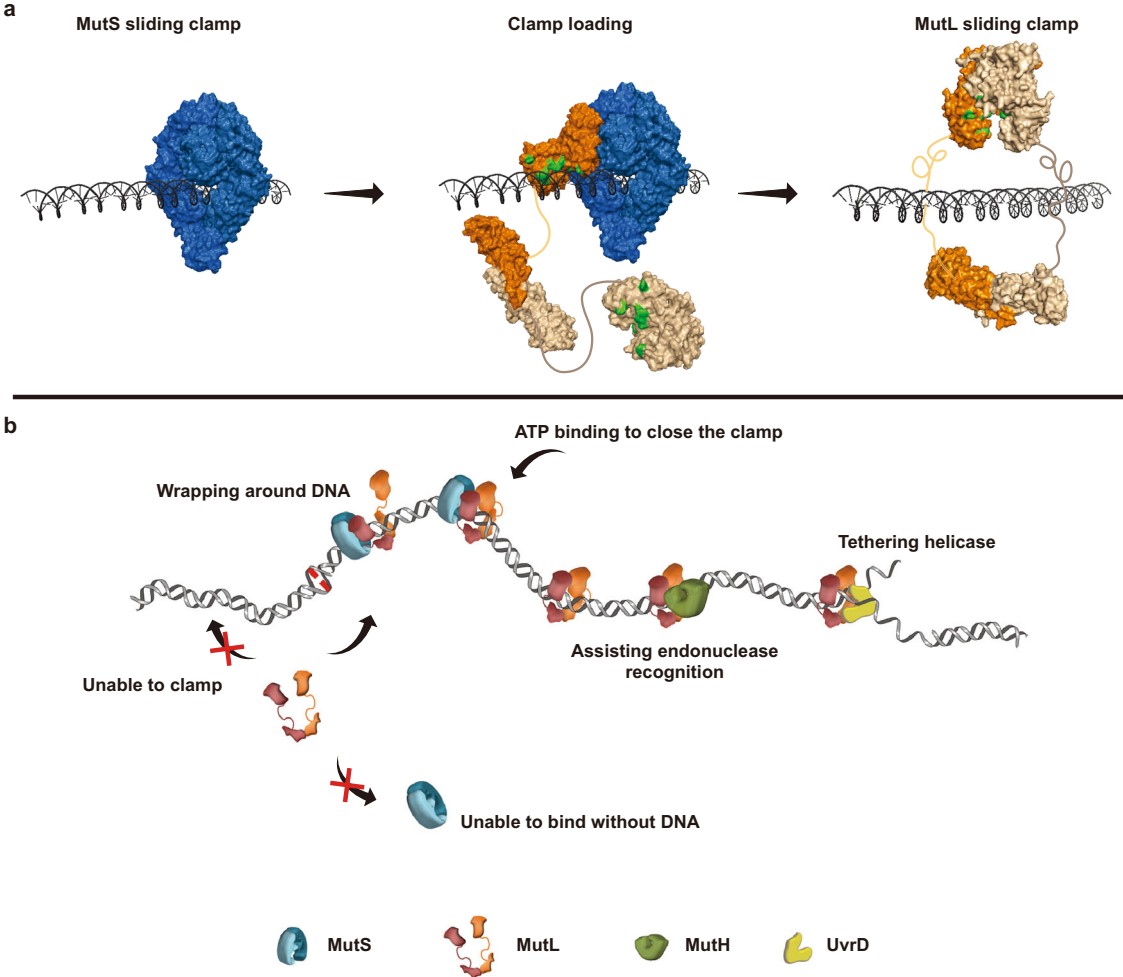

**Fig. 8 | Models for MutL clamp loading by MutS with DNA and functions in MMR. a** Space-filling models of a MutS sliding clamp (left), MutL clamp loading by a MutS (center, PDB ID: 7AIB), and a MutL sliding clamp (right) on DNA. The location of the conserved Arg/Lys residues on MutL PCC are shown in green. Intrinsic disordered linker connecting N- and C-terminal domains of MutL are drawn as curvy lines. **b** A model of the complete *E. coli* MMR process. See text and ref. 20 for detailed descriptions. Shape and color-coded MMR proteins are shown in key below.

or the prospective mechanical functions of a MutL sliding clamp in the multi-step reactions that leads to MMR[13,14]. With the recognition of MutL sliding clamps the potential role(s) for the MutL PCC in MMR were considerably expanded to encompass crucial interaction(s) between MutS and MutL, the material formation of the MutL sliding clamp and/or the distinctive collaborations between the MutL sliding clamp with MutH or UvrD[13,14].

We unmistakably demonstrate that *E.coli* MutL does not bind DNA at physiological salt (Fig. 1c)[50]. As predicted by historical studies, mutations of the conserved MutL PCC Arg/Lys residues progressively reduced a low ionic strength DNA binding activity (Fig. 1c, Supplementary Fig. 3)[23,37,48,49]. Interestingly, we found that MutS could load the mutant PCC MutL proteins as sliding clamps onto a mismatched DNA under ionic conditions that were somewhat higher than their detectable DNA binding activity (Fig. 3d). However, the maximum ionic strength where DNA loading of a PCC-mutant MutL sliding clamp was observed was still well below physiological ionic strength where significant numbers of wild-type MutL can be loaded as a sliding clamp onto the DNA (Fig. 3d). Taken as a whole, these studies correlated MutL sliding clamp formation with DNA engagement by the mutant PCC proteins and demonstrated substantial stimulation of MutL sliding clamp formation in the presence of MutS sliding clamps. Combined with recent structural analysis showing a MutS sliding clamp bound to the MutL NTD that simultaneously positions a remote PCC on the DNA[22,23], these observations are consistent with the conclusion that MutS-MutL complex formation encourages MutL PCC-DNA interactions, which ultimately induce the formation of a MutL sliding clamp. It should be noted that the MutS-MutL-PCC engagement on DNA appears continuously dynamic at the frame rates used in these studies (100 ms), further reducing the likelihood of a static DNA binding complex during MMR processes.

How does MutS employ the PCC during MutL clamp loading? MutL only associates with MutS when the latter is a sliding clamp on the DNA (Fig. 2; Fig. 8b). Moreover, the lifetime of this association is relatively short (-30 s; Fig. 2g)[13]. This observation seems to contrast reports of long-lived affinities between MutS and MutL homologs[64,65]. The available single-molecule imaging and structural evidence suggest that the MutS sliding clamp mechanically positions a single MutL NTD on the DNA, where the PCC may maintain continuous contact with the phosphate backbone (Fig. 1e; Fig. 8a, center)[13,22,23]. Such a localization is likely to instigate the rotation-coupled diffusion of the MutS-MutL complex that has been observed in previous single molecule analysis[13]. We speculate that the rotational diffusion of the MutS-MutL complex aids in the diffusion-mediated wrapping of the remaining MutL domains around the DNA, with ATP binding and dimerization of the NTDs ultimately creating the MutL sliding clamp (Fig. 8a, right). This hypothesis suggests that the capability to wrap MutL protein segments around the DNA would be significantly influenced by the length of the IDL. In support of this concept, progressive linker domain deletions that should gradually reduce diffusion-mediated wrapping efficiency as well as the size of the MLH/PMS donut hole, initially prevent their ability to transit roadblocks on the DNA and eventually completely inhibit MMR[27,60]. However, further studies will be necessary to dissect the detailed role of MLH/PMS IDLs in clamp formation.

We found the MutL PCC is not required for the stable association of the MutH endonuclease with the MutL sliding clamp (Fig. 6c)[13]. However, the MutL PCC appears to be exploited by the MutL-MutH sliding clamp complex to search and incise hemimethylated GATC sites on the DNA (Figs. 6f, 8b). We noted that the conformation of a free MutL NTD may be different from that of a DNA-bound one (Fig. 8a, middle). It is likely that the MutH binds to both configurations but only the latter is capable of interrogating DNA backbone to locate a hemimethylated GATC site. Similarly, the PCC is employed for the colocalization of the MutL sliding clamp with the UvrD helicase at a strand scission (Fig. 7c) as well as for unwinding and release of the mismatch-

containing strand (Figs. 7e, 8b). The simplest interpretation of these observations is that the MutL PCC is engaged by UvrD during nascent DNA unwinding events at a MutH strand scission[66]. Such an engagement would seem to explicitly authorize the MutL sliding clamp as a helicase processivity factor during downstream MMR strand-specific excision[14]. However, a managing role for the MutL PCC in the mechanics of unwinding cannot be ruled out since extensive UvrD helicase association and unwinding may be recovered at intermediate ionic strength (50 mM NaCl) with MutL(R-E) sliding clamps that have been loaded at very low ionic strength by MutS (10 mM NaCl; Supplementary Fig. 6a).

We also demonstrated that MutL contains an intrinsic PCC-dependent DNA clamping activity in low ionic strength conditions (Fig. 3d). This might at least partly explain historical studies that showed UvrD helicase activity could be stimulated by MutL under similar low ionic strength conditions in the absence of a mismatch or MutS[37,49]. Taken together, our studies provide evidence showing the *E. coli* MutL PCC participates in MutL sliding clamp formation, and adds to the previous work[37,48,49] by establishing that the MutL PCC is engaged by downstream MutH endonuclease and UvrD helicase activities in physiological ionic conditions once the MutL protein has formed a sliding clamp on the mismatched DNA. The functional organization of an otherwise undetectable MutL PCC DNA binding activity by multiple protein complexes with clearly different DNA metabolic activities appears to be unique to MMR and represent an attractive biophysical model for understanding the regulation of DNA binding by multiplex protein interactions. Interestingly, while the MLH/PMS PCC residues are conserved among species (Fig. 1a), no MutH or UvrD homologous protein have been found in eukaryotes. Instead, eukaryotic MMR incision relies on interactions between MLH/PMS and PCNA. Further studies will be necessary to explore the roles of MLH/PMS PCC in the clamp formation and downstream eukaryotic MMR events.

It has been generally considered that the MutS and MutL proteins form a stable functional complex to initiate and execute MMR[3,41–43]. However, we found that once a MutL sliding clamp has been loaded onto the mismatched DNA there are no additional requirements for MutS during MutH incision at hemimethylated GATC sites or for the unwinding of the mismatched DNA by UvrD helicase (Figs. 6f, 7e). These observations suggest that the only role for MutS during *E.coli* MMR is as a mismatch-dependent clamp loader of MutL. Our studies strongly suggest that both a MutS sliding clamp and DNA are required to encourage ATP binding-dependent MutL NTD dimerization that results in clamp loading. Conversely, apparently independent ATP hydrolysis by MutL releases the clamp from the DNA (Fig. 5). These cascading MMR sliding clamp progressions are substantially more elementary than the loading and unloading sequence of the well-described replication sliding clamps, β-clamp and PCNA[32,34]. For both β-clamp and PCNA, ATP binding by the clamp loader is employed to form a solution complex between the clamp and clamp loader[32,34], while ATP hydrolysis is used to transfer the sliding clamp to a primer-template[31,32,34]. In eukaryotes, the unloading program swaps at least one protein component from the core clamp loading complex to physically remove PCNA[32,34]. Such widely divergent mechanisms highlight the variety of biophysical solutions for stably loading important genome maintenance proteins onto DNA.

## Methods

### Plasmid construction, MMR protein labeling, and purification
The *E. coli* MutS, MutS-bio, MutL, MutL(R266E), MutL(R162E,R266-E,R316E) [MutL(R-E)], MutH, and UvrD proteins were labeled using sortase-mediated peptide ligation[25]. The MMR genes were amplified by PCR (Supplementary Table 1), digested with XbaI and EcoRI (for MutS), XbaI and BamHI (for MutS-bio), NdeI and XhoI (for MutL), NdeI and BamHI (for MutH) or XbaI and HindIII (for UvrD), and inserted into

pET29a (Novagen) bacterial expression plasmid. Hexa-histidine (his$_6$) and sortase recognition sequence (srt, LPETG) were introduced onto the C-terminus of MutS, MutS-bio, MutL and UvrD proteins, or N-terminus of MutH protein. The avi-tag sequence (GLNDI-FEAQKIEWHE) was introduced between his$_6$ and srt of MutS-bio. The MutL(R266E) and MutL(R-E) mutations were generated using the QuikChange site-directed mutagenesis kit (Stratagene). Two serine residues separated the his$_6$ and srt, and these tags were separated from the MMR proteins by four glycine residues. All the plasmid constructs were amplified in *E. coli* DH5α and verified by DNA sequencing.

After transformation with the MutS, MutH or UvrD expression plasmid, a single colony of BL21 AI cell was diluted into 1 L of LB containing 50 μg/ml kanamycin. At OD$_{600}$ = 0.3, the growth temperature was decreased to 16 °C and the expression of MutS, MutH or UvrD was induced by the addition of L-(+)-Arabinose (0.2% wt/vol) and IPTG (0.2 mM) at 16 °C for 16 h. For the MutL, MutL(R266E) or MutL(R-E) protein, expression was induced by L-(+)-Arabinose (0.2% wt/vol) and IPTG (0.2 mM) at 37 °C for 3 h. For the MutS-bio expression, BL21 AI cell was co-transformed with MutS-bio and BirA expression plasmid (Addgene plasmid #20857) and was grown in LB containing 50 μg/ml kanamycin, ampicillin and 0.05 mM biotin as described[67]. Cells were collected and resuspended in Freezing Buffer (25 mM Hepes pH 7.8, 500 mM NaCl, 10% glycerol, and 20 mM imidazole). Cell pellets were freeze-thawed three times, sonicated twice, and centrifuged at 48,000 × *g* for 1 h. The supernatants were then loaded on a Ni-NTA (Qiagen) column, washed with Buffer A (25 mM Hepes pH 7.8, 500 mM NaCl, 10% glycerol and 20 mM imidazole) and eluted with Buffer B (25 mM Hepes pH 7.8, 500 mM NaCl, 10% glycerol and 200 mM imidazole). Fractions containing MMR proteins were pooled and dialyzed overnight against Labeling Buffer (50 mM Tris-HCl pH 7.8, 150 mM NaCl, 10 mM CaCl$_2$ and 10% glycerol). The protein fractions were then incubated with sortase and Cy3- or Cy5-labeled peptides (GGGC-Cy3/Cy5 for C-terminus labeling and Cy3/Cy5-CLPETGG for N-terminus labeling, purchased from ChinaPeptides Co.,LTD) at 4 °C for 1 h (protein: sortase: peptide in the ratio of 1:2:5). After labeling, MutS, MutS-bio, MutH or UvrD protein was diluted with 2 volumes of Buffer C (25 mM Hepes pH 7.8, 1 mM DTT, 10% glycerol and 0.1 mM EDTA) and loaded onto a heparin columM, washed with Buffer C plus 100 mM NaCl and eluted with Buffer C plus 1 M NaCl. The MutL, MutL(R266E) or MutL (R-E) protein was diluted with 6 volumes of Buffer C and loaded onto a ssDNA cellulose columM, washed with Buffer C plus 25 mM NaCl and eluted with Buffer C plus 0.5 M NaCl. Protein-containing fractions were dialyzed against Storage Buffer (25 mM Hepes pH 7.8, 1 mM DTT, 0.1 mM EDTA, 150 mM NaCl and 20% glycerol) and frozen at −80 °C. The concentrations and labeling efficiencies of MMR proteins were determined by measuring protein absorbance at 280 nm and Cy3/Cy5 absorbance at 550/650 nm, respectively.

## MMR complementation in vivo
*E. coli* strains used in these studies (wild type and *ΔmutL*) were derivatives of MG1655 (F- lambda- *ilvG- rfb*−50 *rph*−1) and were purchased from Guangzhou Ubigene Biosciences Co., Ltd. *ΔmutL* strains were co-transformed with a MutL/MutL(R266E)/MutL(R-E) expression plasmid and pTARA plasmid (for T7 RNA polymerase expression, a gift from Kathleen Matthews, Addgene plasmid #31491). Single colonies were picked and grown for 24 h in the presence of 50 μg/mL Kanamycin, 35 μg/mL Chloramphenicol and 0.2 % Arabinose. As controls, single colonies of wild type and *ΔmutL* strains with pTARA plasmid were grown for 24 h in the presence of 35 μg/mL Chloramphenicol and 0.2% Arabinose. All cell culture samples were first calibrated to identical density (OD$_{600}$ = 2) and dilutions of the cultures were dropped on LB-Agar plates with 100 μg/mL rifampicin. Plates were grown overnight at 37 °C.

## ATPase analysis
The ATPase activity of MutL or MutL(R-E) was measured utilizing an ATPase/GTPase Activity Assay Kit (Sigma). The analysis was carried out with 5 μM protein in a 40 μL reaction mixture comprised of 20 mM Tris-HCl (pH 7.8), 75 mM NaCl, 1 mM ATP, 4 mM MgCl$_2$, and 0.5 mM EDTA. The reactions were performed at 23 °C for 0, 5, 10, 15, and 20 min followed by quenching with 200 μL of malachite green reagents. Samples were incubated at 23 °C for an additional 30 min and transferred to a 96-well plate. Free phosphate was determined by measuring absorbance at 620 nm using a microplate reader Eon (Bio Tek). Data were fit to a linear function to calculate the rates of ATP hydrolysis and the turnover numbers ($k_{cat}$).

## Construction of a circular DNA containing a single mismatch, a strand scission, a Cy3-label, a biotin-label, and a hemi-methylated GATC site
A modified 4.8 kb pRSET-B plasmid (Supplementary Note 1) containing four BbvCI sites was purified from *E. coli* DH5α strain. The plasmid DNA (60 nM) was first digested with Nt.BbvCI (New England Biolabs) to generate four site-specific strand scissions (nicks). Oligonucleotides 7 and 8 (6 μM for each; Supplementary Table 1) were added, the DNA heated to 95 °C for 10 min, cooled slowly to room temperature and ligated (T4 DNA Ligase, New England Biolabs) at 16 °C overnight. The resulting product (Cy3-biotin mismatched DNA; Supplementary Fig. 5a) was separated on a 0.5% agarose gel and the circular DNA was excised and purified using Agarose Gel DNA Extraction Kit.

To confirm the presence of a G/T mismatch, the purified Cy3-biotin DNA was digested simultaneously with a restriction endonuclease that overlaps the mismatch (NcoI or NsiI) and XhoI that linearizes that plasmid (New England Biolabs; Supplementary Fig. 5b). Restriction products were then resolved on a 1% agarose gel. The presence of the biotin label was confirmed by incubating 400 ng circular DNA with 50 μL streptavidin-coated superparamagnetic beads (SA beads, Dynabeads™ MyOne™ C1, Invitrogen) at room temperature for 30 min. The supernatant was collected following precipitation of the biotin-plasmid DNA using a DynaMag™−2 Magnet (Invitrogen). Plasmid DNA remaining in the supernatant was resolved on a 1% agarose gel (Supplementary Fig. 5c).

## MutH endonuclease activity
MutH endonuclease activity was examined on Cy3-biotin mismatched circular DNA prepared as described above, containing wild-type MutS + wild-type MutL, wild-type MutS + MutL(R-E) or wild-type MutL alone. The Cy3-biotin mismatched DNA (800 ng) was first incubated with 100 μL SA beads in Loading Buffer A (20 mM Tris-HCl pH 7.5, 0.1 mM DTT, 5 mM MgCl$_2$, 0.2 mg/mL acetylated BSA) plus 500 mM NaCl at room temperature for 30 min. The Cy3-biotin mismatched DNA bound to SA beads was precipitated and then equilibrated with 200 μL Loading Buffer A plus 100 mM NaCl. For wild-type MutS + wild-type MutL, similar precipitation and equilibration was first performed with wild-type MutS (100 nM) plus 1 mM ATP in Loading Buffer A plus 100 mM NaCl (100 μL). This mixture was incubated at room temperature for 5 min to load MutS sliding clamps onto the Cy3-biotin mismatched DNA, followed by precipitation and equilibration with wild-type MutL (100 nM) in Loading Buffer A plus 100 mM NaCl (100 μL). After 1 min at room temperature, AMP-PNP (1 mM) was added to the nascent assembled MutS-MutL complexes to induce the formation of long-live MutL sliding clamps on the circular Cy3-biotin mismatched DNA (Figs. 5, 6d–f, 7d, e).

MutL(R-E) sliding clamps were loaded onto the Cy3-biotin mismatched DNA by first loading of MutS sliding clamps as described above. The Cy3-biotin mismatched DNA bound to SA beads was then precipitated and equilibrated with MutL(R-E) (100 nM) in Loading Buffer A plus 10 mM NaCl (100 μL). After 1 min at room temperature, AMP-PNP (1 mM) was added to nascent assembled MutS-MutL(R-E)

complexes to induce the formation of MutL(R-E) sliding clamps on the circular Cy3-biotin mismatched DNA (Figs. 6d–f, 7d, e).

Wild-type MutL sliding clamps alone were loaded in the absence of MutS by first incubating the Cy3-biotin mismatched DNA (800 ng) with SA beads in Loading Buffer A plus 100 mM NaCl at room temperature for 5 min. The Cy3-biotin mismatched DNA bound to SA beads was precipitated and equilibrated with MutL (100 nM) in Loading Buffer A plus 10 mM NaCl (100 μL). After 1 min at room temperature, AMP-PNP (1 mM) was added to induce the formation of MutL sliding clamps on the circular Cy3-biotin mismatched DNA.

MutH endonuclease activity was examined with the Cy3-biotin mismatched DNA containing wild-type MutS + wild-type MutL, wild-type MutS + MutL(R-E) or wild-type MutL alone prepared as described by first washed the precipitated DNA-SA beads collected in the last step with 200 μL Loading Buffer A plus 100 mM NaCl three times. The precipitated Cy3-biotin mismatched DNA was then equilibration with MutH (100 nM) and 1 mM ATP in Loading Buffer A plus 100 mM NaCl (100 μL) and incubated at 37 °C for 30 min. The reaction was stopped by the addition of 25 mM EDTA, the supernatant was removed, and the DNA eluted as described previously[68]. Briefly, the SA beads were resuspended in 20 μL ddH₂O and heated to 70 °C at a rate of 0.5 °C per sec. The samples were then cooled to room temperature. The DNA in the supernatant was collected, digested with BbvCI at 37 °C for 1 h, resolved on a 12% denaturing PAGE, scanned on a FLA-9000 imager (FUJIFILM), and quantified by ImageQuant software.

To detect the MutS and MutL sliding clamp, the circular Cy3-biotin mismatched DNA bound to SA beads was precipitated and equilibrated with 200 μL Loading Buffer A plus 100 mM NaCl three times to remove unbound proteins. After 5 min incubation at 4 °C, 40 μL SDS loading buffer (50 mM Tris-HCl pH 6.8, 2% SDS, 10% glycerol, 100 mM DTT) was added and the sample was incubated at 95 °C for 10 min. The SA beads were precipitated and proteins remaining in the supernatant were collected and resolved on 8% SDS-PAGE.

### UvrD helicase unwinding assay
To examine UvrD unwinding activity the circular Cy3-biotin mismatched DNA was first digested with Nt.BbvCI to generate strand scissions flanking the 72 bp DNA fragment containing the mismatch and Cy3-label. Wild-type MutL sliding clamps and MutL(R-E) sliding clamps were loaded onto the circular Cy3-biotin mismatched DNA containing Nt.BbvCI strand scissions as described for MutH endonuclease analysis (Fig. 7d, e). The wild-type MutL sliding clamps alone were loaded onto the circular Cy3-biotin mismatched DNA at very low ionic strength (10 mM NaCl). Precipitated DNA-SA beads were first washed three times with 200 μL Loading Buffer A plus 100 mM NaCl, followed by precipitation and addition of UvrD (50 nM) and 1 mM ATP in Loading Buffer A plus 100 mM NaCl (50 μL). The DNA-SA beads were incubated at 37 °C for 30 min, the reaction was stopped by the addition of 25 mM EDTA, released Cy3-labeled 72 nt DNA in the supernatant collected, resolved on a 12% denatured PAGE, and scanned on a FLA-9000 imager (FUJIFILM) (Fig. 7d, e). Alternatively, the circular Cy3-biotin mismatched DNA containing Nt.BbvCI strand scissions may be incubated directly with MutS (100 nM), MutL (100 nM), and UvrD (50 nM) with 1 mM ATP in Loading Buffer A plus 100 mM NaCl at 37 °C for 30 min, the reaction stopped by the addition of 25 mM EDTA, the DNA resolved on a 1% Agarose gel, and scanned on a FLA-9000 imager (FUJIFILM). Loss of the Cy3-label from the circular Cy3-biotin mismatched DNA is indicative of helicase unwinding and release (Supplementary Fig. 6b, c).

### Construction of 18.4-kb λ phage-based DNA with a single mismatch
The mismatched DNA was prepared as described previously[14]. A plasmid containing two BsaI sites was first treated with BsaI (New England Biolabs), then separated on a 1% agarose gel. The 7-kb band was

excised and recycled using Agarose Gel DNA Extraction Kit (TaKaRa Bio). Concurrently, λ phage DNA (3.2 nM, Thermo Fisher Scientific) was ligated with oligo 1 and oligo 2 (800 nM; Supplementary Table 1) at room temperature (23 °C) overnight. Unligated oligonucleotides were removed using a 100 kDa Amicon filter (Millipore). The resulting λ DNA was then digested with BsaI at 37 °C for 3 h, ligated with the 7-kb DNA, 1000 × oligo 3 and oligo 4 (Supplementary Table 1) at 18 °C overnight. DNA ligation products were separated on a 0.5% low melting agarose (Promega) gel and the 18.4-kb band was excised and treated with β-Agarase (Sigma) followed by isopropanol precipitation. The purified DNA was resuspended in TE buffer (10 mM Tris-HCl, pH 7.5, 1 mM EDTA) and stored at −80 °C until use.

### Single-molecule imaging buffers and experiment conditions
The single-molecule Imaging Buffer A contains 20 mM Tris-HCl (pH 7.5), 0.1 mM DTT, 0.2 mg/mL acetylated BSA (Molecular Cloning Laboratories), 0.0025% P-20 surfactant (GE healthcare), 1 mM ATP, 5 mM MgCl₂ (unless stated otherwise) and 100 mM NaCl (unless stated otherwise). To minimize photoblinking and photobleaching, All Imaging Buffer was supplemented with a photostability enhancing and oxygen scavenging cocktail containing saturated (-3 mM) Trolox and PCA/PCD oxygen scavenger system composed of PCA (1 mM) and PCD (10 nM)[69].

### Single-molecule total internal reflection fluorescence (smTIRF) microscopy
All the single-molecule total internal reflection fluorescence (smTIRF) data were acquired on a custom-built prism-type TIRF microscope established on the Olympus microscope body IX73. Fluorophores were excited using the 532 nm for green and 637 nm for red laser lines built into the smTIRF system. Image acquisition was performed using an EMCCD camera (iXon Ultra 897, Andor) after splitting emissions by an optical setup (OptoSplit II emission image splitter, Cairn Research). Micro-Manager image capture software was used to control the opening and closing of a shutter, which in turn controlled the laser excitation.

The 18.4-kb mismatched DNA (300 pM) in 300 μL T50 buffer (20 mM Tris- HCl, pH 7.5, 50 mM NaCl) was injected into a custom-made flow cell chamber and stretched by laminar flow (300 μL/min). The stretched DNA was anchored at both ends onto a neutravidin-coated, PEG passivated quartz slide surface, and the unbound DNA was flushed by similar laminar flow.

To examine the MutL DNA binding activity and lifetime, 2 nM Cy3-MutL in Imaging Buffer A (plus 10–60 mM NaCl and 0 mM MgCl₂) was introduced into the flow cell chamber. To determine the diffusion coefficients (D) of MutL molecule on DNA, Cy3-MutL (1–20 nM) in Imaging Buffer A (10–50 mM NaCl and 0 mM MgCl₂) was introduced into the flow cell chamber. The interactions between MutL and DNA were monitored in real-time in the absence of flow at ambient temperature (-23 °C). The DNA was located by stained with Sytox orange (250 nM, Invitrogen) after real-time recording.

To examine the MutS-MutL interactions on mismatched DNA, Cy5-MutS (3 nM) and Cy3-MutL (10 nM) in Imaging Buffer A were introduced into the flow cell chamber and protein-protein interactions were monitored in real-time in the absence of flow at ambient temperature (-23 °C). To examine the MutS-MutL(R-E) interactions on mismatched DNA under low ionic strength, Cy5-MutS (5 nM) and MutL(R-E)-Cy3 (20 nM) in Imaging Buffer B (20 mM Tris-HCl pH 7.5, 0.1 mM DTT, 0.2 mg/mL acetylated BSA, 0.0025% P-20 surfactant, 1 mM ATP, 1 mM MgCl₂ and 10 mM NaCl) were introduced into the flow cell chamber and protein–protein interactions monitored in real-time.

To examine the MutS-MutL interactions with a short blocked-end DNA substrate a 59-bp mismatched DNA was first constructed by annealing two oligos (oligo 5 and oligo 6; Supplementary Table 1). Cy5-bio-MutS was immobilized on the PEG and PEG-neutravidin passivated

quartz surface, followed by the injection of 10 nM Cy3-MutL, 100 nM 59-bp mismatched DNA containing 5′-biotin on both ends, and 5 μM neutravidin in Imaging Buffer A. Protein co-localizations were monitored in real-time in the absence of flow.

To examine the formation of the MutL ring-like clamp on mismatched DNA, Cy5-MutS (3 nM), and Cy3-MutL (10 nM) in Imaging Buffer A were introduced into the flow cell chamber and fast-diffusing MutL molecules were monitored in real-time in the absence of flow. To examine the formation of MutL sliding clamp by buffer exchange, Cy5-MutS (3 nM) and Cy3-MutL (10 nM) in Imaging Buffer A (plus 10–100 mM NaCl and 1 mM MgCl$_2$) were first introduced into the flow cell chamber. After 5 min, the flow cell was flushed with Imaging Buffer A, and fast-diffusing MutL molecules were monitored in real-time in the absence of flow. To examine the MutS-MutL complex formed by MutS and MutL clamps, Cy5-MutS and Cy3-MutL in Imaging Buffer B were first introduced into the flow cell chamber. After 5 min, the flow cell was flushed with 1 nM Cy5-MutS in Imaging Buffer A, and protein-protein interactions were monitored in real-time in the absence of flow.

To examine the formation of ATP/AMP-PNP-bound MutL clamps on mismatched DNA, MutS (unlabeled, 20 nM) in Imaging Buffer A was first introduced into the flow cell chamber. After 5 min incubation, the flow cell was flushed with Cy3-MutL (80 nM) in Imaging Buffer C (20 mM Tris-HCl pH 7.5, 0.1 mM DTT, 0.2 mg/mL acetylated BSA, 0.0025% P-20 surfactant, 0 mM ATP, 5 mM MgCl$_2$ and 100 mM NaCl). After 1 min, the flow cell was flushed with 1 mM ATP or AMP-PNP in Imaging Buffer C. The ATP/AMP-PNP-bound MutL molecules were then monitored in real-time in the absence of flow.

To examine the interactions between the MutL sliding clamp and MutH endonuclease, MutS (unlabeled, 10 nM), Cy3-MutL (20 nM) and MutH-Cy5 (10 nM) in Imaging Buffer A were introduced into the flow cell chamber and protein-protein interactions were monitored in real-time in the absence of flow. To measure the interactions between the MutL sliding clamp and UvrD helicase, MutS (unlabeled, 100 nM), MutL (100 nM), and UvrD-Cy5 (30 nM) in Imaging Buffer A were introduced into the flow cell chamber and protein–protein interactions were monitored in real-time in the absence of flow. To examine the interactions between MutL(R-E) clamp and MutH/UvrD by buffer exchange, MutS (unlabeled, 50 nM), and MutL(R-E)-Cy3 (50 nM) in Imaging Buffer B were first introduced into the flow cell chamber. After 5 min, the flow cell was flushed with MutH-Cy5 (20 nM) or UvrD-Cy5 (50 nM) in Imaging Buffer A and protein-protein interactions were monitored in real-time in the absence of flow.

To measure the interactions between AMP-PNP-bound MutL and MutH, MutS (unlabeled, 20 nM) in Imaging Buffer A was first introduced into the flow cell chamber. After 5 min incubation, the flow cell was flushed with Cy3-MutL (80 nM) in Imaging Buffer C. After 1 min, the flow cell was flushed with 10 nM MutH-Cy5 plus 1 mM AMP-PNP in Imaging Buffer C. To examine the interactions between AMP-PNP-bound MutL and UvrD, MutS (unlabeled, 100 nM) in Imaging Buffer A was first introduced into the flow cell chamber. After 5 min, the flow cell was flushed with Cy3-MutL (100 nM) in Imaging Buffer C. Then after 1 min, the flow cell was flushed with 1 mM AMP-PNP in Imaging Buffer C. Finally, after 5 min, the flow cell was flushed with UvrD-Cy5 (30 nM) in Imaging Buffer A and protein-protein interactions were monitored in real-time in the absence of flow.

### Data analysis of TIRF imaging

To determine the starting positions of MutS or MutL on DNA, the 18.4-kb mismatched DNA was stained with Syto 59 (700 nM, Invitrogen) or Sytox Orange (250 nM, Invitrogen). The left (P$_L$) and the right (P$_R$) end positions of the DNA as well as the horizontal positions of diffusing particles (P$_P$) along the DNA were determined as previously described[13]. The positions were then converted to lengths in bp by the following equation: 18,378 bp × (P$_P$ − P$_L$)/(P$_R$ − P$_L$), where 18,378 bp is the length of the mismatched DNA. A 1000 bp (~2 pixels) binning size was used to construct the position histograms.

To determine the diffusion coefficient of MutL, the MutS-MutL/MutL(R-E) complex and the MutL/MutL(R-E) clamps, particles were tracked using DiaTrack 3.04 to obtain single-molecule trajectories. Diffusion coefficients were calculated from the trajectories as previously described[13]. Briefly, the diffusion coefficient (D) was determined from the slope of a mean-square displacement (MSD) versus time plot using the equation MSD(t) = 2 Dt, where t is the time interval. The first 10% of the total measurement time was taken for point fitting. A minimum number of 50 frames were used to calculate the diffusion coefficients.

A 100–300 ms frame rate was used to examine MutL, MutS-MutL, MutL-MutH or MutL-UvrD complex on DNA. To determine the survival probability of the MutL/MutL(R-E) clamp on DNA, a 3000-ms or 6000-ms frame rate with 300-ms laser exposure time was used to minimize photo-bleaching. To determine the dwell time of MutS-MutL interaction on short DNA substrate, a 2000-ms frame rate with 300-ms laser exposure time was used to minimize photo-bleaching. Kymographs were generated along the DNA by a kymograph plugin in ImageJ (J. Rietdorf and A. Seitz, EMBL Heidelberg). To plot the survival probability of MutL/MutL(R-E) clamp on DNA, the number of MutL clamp at the beginning of each movie was set to 1, and MutL dissociation was quantified in 60-s or 300-s time bins.

To determine the frequency of MutL on DNA, single-molecule movies were recorded for 2 min and the diffusing MutL molecules with a minimum lifetime of 3 s were counted as the number of MutL (N$_L$). To measure the frequency of the MutS-MutL complex on DNA, single-molecule movies were recorded for 12 min. Cy3 and Cy5 channels were merged and co-localized molecules with a minimum lifetime of 10 s were counted as the number of MutS-MutL complex (N$_{SL}$). To determine the frequency of MutL-MutH complex on DNA, single molecule movies were recorded for 12 min. Cy5-MutH molecules with a minimum lifetime of 10 s were counted as the number of MutL-MutH complex (N$_{LH}$). To determine the frequency of the MutL-UvrD complex on DNA, single-molecule movies were recorded for 12 min. Cy5-UvrD molecules with a minimum lifetime of 10 s and a minimum DNA movement of 333 nm (2 pixels, unidirectionally) were counted as MutL-UvrD complex (N$_{LU}$). To determine the frequency of the MutL clamps on DNA, single-molecule movies were recorded for 12 min. To determine the frequency of ATP/AMP-PNP-bound MutL clamp on DNA, single-molecule movies were recorded for 30 min. MutL molecules with a minimum lifetime of 30 s and a minimum diffusion coefficient of 0.1 μm$^2$ s$^{-1}$ were counted as the number of MutL clamps (N$_{L\text{-clamp}}$). We only included molecules that were clearly diffusing/moving along the DNA (>5 frames) in the analysis. Following the real-time single-molecule recording, the number of DNA molecules (N$_{DNA}$) was determined by Sytox Orange staining. The frequencies of MutL (F$_L$), MutS-MutL complex (F$_{SL}$), MutL-MutH complex (F$_{LH}$), MutL-UvrD complex (F$_{LU}$) and MutL clamp (F$_{L\text{-clamp}}$) were calculated using the following equations that also included corrections for labeling efficiencies of the proteins (the numbers in the denominator, Supplementary Table 2):

$$F_L = \frac{N_L}{N_{DNA} \times 0.71} \tag{1}$$

$$F_{SL} = \frac{N_{SL}}{N_{DNA} \times 0.71 \times 0.8} \tag{2}$$

$$F_{LH} = \frac{N_{LH}}{N_{DNA} \times 0.9} \tag{3}$$

$$F_{\text{LU}} = \frac{N_{\text{LU}}}{N_{\text{DNA}}} \qquad (4)$$

$$F_{\text{L-clamp}} = \frac{N_{\text{L-clamp}}}{N_{DNA} \times 0.71} \qquad (5)$$

To determine the frequency of MutS-MutL complex in co-localization studies, single-molecule movies were recorded for 10 min. MutS and MutL molecules were tracked by SPARTAN to generate trajectories[70]. Co-localized molecules with a lifetime between 10 and 200 s were counted as MutS-MutL complex ($N_{\text{SL-co}}$). The total numbers of MutS trajectories were counted as the number of Cy5-bio-MutS molecules ($N_{\text{S}}$). The frequencies were calculated using the following equations that also included corrections for labeling efficiencies of the proteins (the numbers in the denominator, Supplementary Table 2):

$$F_{\text{L}} = \frac{N_{\text{SL-co}}}{N_{\text{S}} \times 0.71} \qquad (6)$$

All single-molecule frequency studies were performed at least two separate times.

**Binning method**
All binned histograms were produced by automatically splitting the data range into bins of equal size by using the Origin program.

**Statistics and reproducibility**
All representative experiments (Figs. 6e, 7e; Supplementary Figs. 2b, 4c, 5b, c, e, f) were repeated independently for at least twice with similar results.

**Reporting summary**
Further information on research design is available in the Nature Research Reporting Summary linked to this article.

## Data availability
A full list of oligonucleotides and PCR primers is available in Supplementary Table 1. The raw data support the findings of this study and uncropped images of gels are available as Source Data. Source data are provided with this paper.

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

## Acknowledgements

We would like to thank Dr. Jong-Bong Lee (Pohang University of Science and Technology) for many helpful insights and discussions, and Dr. Shao-Qing Zhang (CAS Center for Excellence in Molecular Cell Science) and Dr. Zhuo Yang (Chemical Biology Core Facility of CAS Center for Excellence in Molecular Cell Science) for technical support. This work was supported by the National Key R&D Program of China (2021YFA1300503), the Chinese Academy of Sciences (CAS) (YSBR-009), the National Natural Science Foundation of China (32071283), the Shanghai Municipal Commission for Science and Technology (20JC1410300), the Natural Science Foundation of Shanghai (20ZR1474100) and the Shanghai Pujiang Program (20PJ1414500) to J. Liu; and NIH grants CA067007 and GM129764 to R.F.

## Author contributions

R.F. and J. Liu conceived and supervised this study. X.-W.Y., J. London, R.F., and J. Liu designed the experiments; X.-W.Y. and X.-P.H., performed

the genetic analysis; X.-W.Y. and C.H. purified and labeled the proteins; X.-W.Y. and X.-P.H. performed the single-molecule and bulk studies; X.-W.Y., J. London, R.F., and J. Liu analysed the data; X.-W.Y., J. London, R.F., and J. Liu wrote the paper. All authors participated in critical discussions.

## Competing interests

The authors declare no competing interests.
