## [Peer Review File · Nature Communications]

MutS Functions as a Clamp Loader by Positioning MutL on the DNA during Mismatch RepairReviewer #1 (Remarks to the Author):

In this manuscript the authors describe how a positively charged patch (PCC) on the N-terminal domain of the mismatch repair protein MutL is involved in loading the MutL sliding clamp on DNA at physiological conditions. The authors initiate their studies with MutL alone and show that DNA binding at physiological salt cannot be detected but becomes apparent at low salt and requires the PCC. The authors then continue with MutS MutL complex formation to show that the PCC is required for MutS-dependent MutL sliding clamp formation at physiological salt but again that at low salt the complex can also be formed by the PCC mutant. The buffer exchanges to switch ionic strength and nucleotide conditions are nice implementations that provide extra information on the behavior of the MutL mutant.

I have two major concerns:

1) There are many studies in the literature about the exact same MutL mutant R162E/R266E/R316E and the single R266E MutL mutant. The study by Junop 2003 is not mentioned at all and should be included since it is the first to report that R266E has a mutator phenotype. Furthermore, several papers that are mentioned in relation to DNA binding and mutator phenotype (Robertson, Guarne, Groothuizen), also contain functional data (MutH activation, helicase activation, protein complex formation) which is not mentioned at all or discussed by the authors. Yet they claim that the role, mechanism of action and importance of the PCC remains unknown. I disagree, because the role, mechanism and importance of the PCC are to a large extent explained by the published data that is (partially) ignored.

2) The authors do not refer to Fernandez-Leiro et al, Nature Structure and Molecular Biology, April 2021, doi: 10.1038/s41594-021-00577-7. In this study, the cryo-EM structure of MutS/MutL complex on DNA is presented, with the dsDNA visible (unlike the Groothuizen 2015 study). It also clearly shows that from the 3 residues that are studied in MutL(R-E), two are indeed contacting the DNA but the third (R316) is too far away to interact with DNA in this conformation. I realize work by the authors on MutL(R-E) was initiated long before Fernandez-Leiro 2021 was published, nevertheless the structure is relevant for this study and has to be referred to at multiple locations in the manuscript. This can all be fixed, by including the missing references and properly citing the observations that have been described in the literature since 2003 at all required locations in the text, proper discussion of current findings, combined with removing the statements about the function of the PCC being unclear/unknown/unresolved. However, when this is included and adapted, it becomes clear that much was known already about the role and importance of the positively charged patch in MutL and that new insights provided by this work are limited to the patch not only being involved in placing MutL correctly on DNA (which was already shown in structural studies) but also in the subsequent step of sliding clamp formation. The importance of the PCC mutant's (in)ability to bind downstream components when put on DNA as sliding clamp at low ionic strength remains an open question as the DNA (methylated and unnicked) does not allow functional analysis of action on DNA (MutH activation, helicase activation).

Detailed explanation of major issue concerning uncited and cited literature:

The authors state in the introduction that the biological function of the PCC remains an enigma, in the first paragraph of the results that the mechanical function(s) and physiological significance of the PCC is unknown, in the fourth paragraph of the results that a physiological role for the MutL PCC in MMR remained uncertain, and in the discussion that the physiological role of the PCC is uncertain at best. This is not correct. There is a large body of information on the R266E single mutant and the R162E/R266E/R316E (MutL(R-E) mutant) in the published literature:

- Junop 2003: R266E has reduced DNA binding and is an in vivo mutator
- Guarne 2004: R266E has reduced DNA binding and reduced UvrD activation.
- Robertson 2006: R266E still interacts with MutH and UvrD, shows reduced nicking by MutH, much reduced UvrD unwinding and has strong in vivo mutator phenotype.
- Groothuizen 2015: R266E partially defective in nicking, MutL(R-E) fully defective in nicking, both mutants strong mutators in vivo. Residues not involved in binding MutS.
- Fernandez-Leiro 2021: CryoEM structure of MutS-MutL NTD-DNA complex shows

interaction of R266 and R162 with DNA

Taken together there is no doubt about the function (mechanical and biological) of the PCC; it is binding DNA, not involved in binding directly to MutS, MutH and UvrD, required for MutH and UvrD activation, and absolutely required for biological function (mutation avoidance). The authors should mention the functional data from all these studies at all relevant moments in the text and rephrase their statements about the role of the PCC so as to be much more specific on what is already known and what not. Especially in the discussion the reported findings about MutH and UvrD binding have to be put in context of the published functional assays.

Other remarks:

- The abstract mentions 'once in a sliding clamp form, the MutL PCC aids in UvrD helicase capture but not interactions with MutH during mismatch excision'. There is however no excision date in this manuscript, so 'during mismatch excision' should be removed.
- First alinea introduction: 'proteins mediate downstream excision events that result in the discrimination of the error containing strand'. This should be the other way around: the proteins first have to discriminate between correct and error-containing strand, only after that they incise and/or excise the error containing strand.
- Second alinea of introduction on page 3: Together with the crystal structure (ref29) the cryoEM structure of Fernandez-Leiro from 2021 is very relevant and should be mentioned.
- Bottom page 4: mutations of several PCC residues within this cleft resulted in impaired MMR -> include Junop et al 2003 in which the mutator phenotype of R226E is already studied, preceding the other studies .
- Page 5 last line first result paragraph (however, the mechanical...): this is one of the sentences about known function of the PCC that should be removed/rephrased (see above).
- Page 6, second result paragraph top: specify in text that this first experiment was performed at physiological salt, makes it easier to follow the logic without having to check in M&M.
- Page 7 top last sentence: this is one more of the sentences that should be rephrased because the author's own mutator assay just confirmed that the MutL PCC has crucial physiological role instead of uncertain physiological role.
- Page 7 new section: First sentence mentions that a MutL activity may be disrupted, this seems to ignore that the authors just have shown in the preceding section that binding to DNA is disrupted. Rephrase more logical.
- Page 9 last sentence upper paragraph: 'it seems likely' should be rephrased because the Fernandez-Leiro 2021 reference (should be included) shows exactly that.
- Page 11 at bottom: "These results suggest that the RE mutant is unable to mediate communications between mismatch recognition and strand incision/excision processes" -> acknowledge here (and in the discussion!) that this was shown before for triple RE mutant and the single R266 mutant in a more direct manner by functional analysis using incision and unwinding/excision assays in multiple studies (Junop 2003, Guarne 2004, Robertson 2006, Groothuizen 2015).
- Page 12: "In contrast, few if any UvrD unwinding events were observed (Fig6E)". According M&M the DNA does not contain a nick. Unwinding could therefore never occur in this setup. It could be the case that the initial MutL-UvrD interaction on its own is too weak to observe in this assay if the UvrD cannot properly engage the DNA, but that proper UvrD binding will be possible in the presence of a nick. It would therefore be interesting to repeat this assay on nicked DNA to see if UvrD in that case can actually be loaded/activated by the MutL(R-E) mutant using the low salt/high salt switch.
- Page 13 almost at bottom: here another of the sentences that should be rephrased: ...DNA binding activity makes the physiological role of the PCC uncertain at best.
- Page 14; The first sentence is not correct. Ref 66 Schofield 2001 and Ref 67 Qiu 2015 do not report on affinities between MutS and MutL independent of DNA. All experiments were performed in the presence of DNA.
- Page 14 second sentence: 'MutS-DNA appears to position a single MutL NTD PCC in

continuous contact with the DNA backbone'. Here Fernandez-Leiro has to be included because that provides direct evidence for 2 residues in the PCC to contact DNA (and the 3rd not).

- Page 14 2nd paragraph 2nd sentence: rephrase to distinguish between the two following possible interpretations: The MutL sliding clamp recognizes the unwinding event created by captured UvrD, or: the MutL creates the nascent unwinding event so that UvrD can be captured (more clearly formulated in Ref20)
- Page 14 2nd paragraph: the authors probably mean 'productive UvrD binding+unwinding' rather than capture (=initial binding) of UvrD.
- Page 14 2nd paragraph last sentence: Include Fernandez-Leiro 2021 reference.
- Page 18: Why was MutL binding studied without magnesium? 1) Magnesium is a relevant counter ion for DNA and influences binding. 2) Is MutL actually capable of binding ATP in the absence of magnesium? How can these results be compared to later results with magnesium? What results are obtained when magnesium is included?
- Page 21: M&M contains docking analysis but nowhere in the text is this referred to. Was it used for figure 7? It is better to use Fernandez-Leiro cryoEM structure to create Figure 7.
- Figure 7 and Figure S1 use the wrong MutL C-terminal subunit configuration. See for example <https://doi.org/10.1016/j.jmb.2005.06.044> Please correct
- Please include information about ATP presence and concentrations in the ATPase assay description in M&M.
- It would be very relevant to include in Figure 1 a close up of DNA binding of MutL to the DNA using the cryoEM MutS-MutL on DNA structure from Fernandez-Leiro 2021.
- Legend Figure 2: would be helpful to include ionic strength for all panels.
- Legend Figure 4: which ionic strength?
- Legend Sup Figure 4: standard deviation on three independent experiment does not mean much because sd is a Gaussian distribution and cannot be properly described by only 3 points. It would be better to choose a different error measure that is informative. Check and correct for complete manuscript.
- How was labeling efficiency determined?
- How was verified that labeling does not interfere with/change any function of the proteins?

1) Minor points:

- Please correct typo/grammar in abstract 'mismatch nucleotides' and 'these studies underlining'
- Last alinea page 3: rephrase 'Dam methylation is utilized to discriminate between parental and the error-containing strand'
- Correct grammar page 4 'where the clamp loader eventually transferring the clamp'
- The title of the first results paragraph says that the PCC is required for MMR, which is correct, but this observation is only a single sentence in this results section (and it was already shown in literature by many other studies). The main body of this result section is DNA binding by wtMutL and PCC mutant at different ionic strengths, this is the new information and I would suggest to rephrase the title of this section to reflect that.
- Page 5 first sentence second paragraph: rephrase 'to visualize E.coli MutL PCC' into something like 'to study the role of the E.coli MutL PCC'.
- What is meant by 'genetic defect' on page 7 first line new section: The mutations from arginine to glutamate? Because they may indeed disrupt MutL activities, but better to write 'mutations'. Or does this refer to the mutator phenotype? Then the grammar should be changed because the phenotype is caused by changes in MutL activity rather than the other way around.
- Same paragraph: the use of 'ensemble' is confusing as this is a single molecule assay and 'ensemble' usually refers to bulk assay.
- Same sentence: to examine MMR -> the assay does not examine the full MMR reaction, but initial steps.
- Page 11 lower paragraph "previous studies...": MutH does not perform excision but incision
- Next sentence: what is meant by "ensemble"? (see above)

- Page 13 bottom: MutL associates with MutS when it is a sliding clamp: does 'it' refer to MutS or MutL?
- Page 14 line 6: rephrase 'MutL peptides' to 'regions' or 'domains' of MutL
- Page 15 line 4: clarify if a protein or a nucleotide (ADP/ATP) or some other component is swapped.
- Legend Sup figure 4: Reverse order: A linear function was fit to the data.
- Check all figures and correct Time (m) -> Time (minutes) (not meter).
- Consider replacing the 0's (zeros) in Table S5 with 'not relevant'

Reviewer #2 (Remarks to the Author):

NCOMMS-21-21036-Z Yang et al

In this paper, the authors use a diversity of single molecule approaches to examine wild-type MutL (MutL) and a version of MutL (MutL (R-E)) containing 3 amino acid substitutions in a previously identified basic DNA binding cleft called the PCC (positively charged cleft). They examine the ability of these proteins to bind DNA and load onto DNA as dynamic sliding clamps where loading is catalyzed by MutS in the presence of a mismatch or loaded at lower rates in the absence of MutS; as part of these studies, they monitor MutL clamp association with MutS clamps and define the key kinetic parameters of these interactions. They also examine the ability of MutS-MutL and MutS-MutL (R-E) to interact with other MMR proteins like MutH and UvrD. Some of the experiments involving experimental changes of buffer to modulate ionic strength to facilitate MutL (R-E) loading at low ionic strength followed by monitoring various downstream properties at higher ionic strength. Overall, the experiments appear to be very well done, and they are described in an extensive methods section. Furthermore, with the exception of the figure legends as commented on further below, the paper is well written.

The studies establish a number of results. First, MutL binds DNA at low ionic strength and not at physiological ionic strength whereas the MutL (R-E) protein can only bind DNA at even further reduced ionic strength. These results, which are established for the first time using single molecule methods, are consistent with previous publications. They show that MutL (R-E) has wild-type ATPase activity. They show that unlike for MutL, MutS cannot load MutL (R-E) onto DNA to form dynamic MutL sliding clamps at physiological ionic strength but can do so at much reduced ionic strength; when these latter mutant clamp complexes are transferred to physiological ionic strength they behave in regard to being a sliding clamp indistinguishably from wild-type MutL clamps with the exception of a shorter lifetime on DNA arguing that the PCC is required for MutL loading. This is the first evidence for a biochemical role for the PCC in MMR, which is important. They also find the wild-type MutL can form sliding clamps at reduced rates without MutS, thus for the first time establishing MutS as a catalyst; I think this result will significantly influence how we think about MMR mechanisms in the future. The authors also provide evidence that ATP hydrolysis by MutL may play a role in turnover of MutL sliding clamps. Finally, they show that the PCC mutant protein once loaded as a sliding clamp at low ionic strength and transferred to physiological ionic strength can like wild type MutL load MutH but unlike wild type MutL cannot load UvrD thus establishing a role for the PCC region in mediating UvrD-MutL interactions during MMR. These results establish many new details of MMR mechanisms and provide new mechanistic insights into the function of a critical surface on the MutL protein.

I feel the experimental observations presented are well established and provide new mechanistic details about MMR. I am concerned about the author's interpretation that the PCC acts in MMR through its interactions with DNA. The authors clearly show that wild type MutL interacts with DNA at low ionic strength and that mutating the PCC weakens this interaction; this is a point in favor of a PCC-DNA interaction. In contrast, the authors show that every feature of MutL action tested (Loading by MutS, formation of clamps with and without MutS, and recruitment on MutH and UvrD) can all occur at

physiological ionic strength where the interaction between the PCC at DNA never occurred; this is a point against a PCC DNA interaction. They show that once MutL (R-E) is loaded on DNA at low ionic strength and transferred to physiological ionic strength where no PCC DNA binding occurs normal MutH binding occurs but UvrD binding is defective; this is a point against a PCC DNA interaction and further how could the postulated PCC interaction with a UvrD displaced single strand occur at physiological ionic strength. The authors basically argue that PCC DNA interactions are important because lowering the ionic strength both enhances MutL (R-E) DNA binding and MutS recruitment, which could be unrelated. Isn't a more logical interpretation that the PCC R-E mutation causes defects in MutL interactions with both MutS and UvrD with the MutS interaction defect being suppressed at low ionic strength and the effect of low ionic strength of UvrD recruitment as yet to be tested. Enhanced DNA binding of MutL (R-E) could play a role in suppression or it could play no role at all. The authors need to present a more open-minded model that better accounts for the data; indeed, the model I suggest is more novel than the author's. Finally, have the authors used MutS to load MutL (R-E) at low ionic strength and then monitor recruitment of MutH and UvrD at low ionic strength, which might be insightful.

As a minor note, I think the figure legends are too abbreviated. This forces the reader to constantly refer to the methods section to understand each experiment, which is a distraction. In addition, some of the kymographs are very dim and either better exposures or larger images would help.

Some specific comments

P5 and Figure 1a. I think figure 1a would be more informative if it also showed the basic patch in a MutL N-terminal dimer so that one can better see the positioning of the basic patch. In addition, the positions of the amino acid substitutions used should be shown.

P6 and Supp Figure 2. It would be helpful if the legend for panels d, e and f contained some additional information about each experiment (what was added, ionic strength, etc) and also what defines the starting position (presumably first visible fluorescence). In general, all of the legends should be looked at from this perspective.

P6, L11. DNA are either

P6, L16-18. Do the authors really have data that allows concluding ion shielding and rotational coupled diffusion?

P7, L1. An elevated frequency of ...

P7, L6, 7. The genetics establish a physiological role. What they don't establish a mechanism underlying this role.

P7, L12. Found a small difference, not very little difference.

P7, L22. Aren't the authors substituting MutL (R-E) for wild-type MutL?

Figure 2e legend. What ionic strengths were used in the two experiments (are they the same or different given the lack of recruitment of DE at physiological ionic strength).

P9, L8-11. It would be nice to have the 2 sites indicated on a structure to support this statement. Possible Supp Figure 1.

P13, L9. Dynamic motion on DNA. Aren't there some SPR references that should be cited?

P14, L21. Mechanical, not mechanically.

Legend to Figure 7b. This legend is truly inadequate. Most of the proteins in this figure panel are not even identified.

Reviewer #3 (Remarks to the Author):

Yang et al. have performed single molecule fluorescence and bulk biochemical measurements to study different aspects of mismatch repair (MMR) mechanism using E. coli MutS and MutL, in addition to UvrD and MthH. In particular, they focused on identifying the role of a conserved positively charged cleft (PCC) located on the MutL N-terminal domain. Mutations in several residues in PCC (R162E, R266E and R316E) were known to result in impaired MMR (ref 29) and much lower DNA affinity at low ionic strengths. Therefore, PCC was considered to be involved with DNA interactions at low ionic strengths.

Based on earlier work of the authors and other researchers in the field, it is generally accepted that MutS identifies mismatched DNA and recruits MutL. MutL is known not to bind to DNA at physiological ionic conditions (although it has non-specific binding at low ionic strengths) but can be recruited to mismatch site by MutS. MMR mechanism and how different proteins involved in MMR interact with each other have been extensively studied. In a recent paper, the authors studied the role of MutL in recruiting UvrD to a mismatch site and the impact of MutL on UvrD processivity (Liu et al. Nat Com 2019) using methods and DNA constructs (18.4 kb DNA with a single mismatch) similar to those employed in this study. These methods are well-established, and the authors are experienced in these assays.

The current study adds to this literature by investigating how the mentioned R-E mutations in PCC impact DNA binding activity of MutL and its interactions with MutS, UvrD and MthH at different ionic strengths and different nucleotide conditions (ATP vs AMP-PNP).

The authors show that, MutL(R-E) could not be recruited to DNA by MutS at physiological ionic conditions even though recruitment events have been observed at very low ionic strengths (10 mM NaCl) in the presence of ATP. Once recruited to DNA at these low ionic strengths, MutS-MutL and MutS-MutL(R-E) have very similar diffusion coefficients. The authors conclude the interactions between PCC and DNA are important for complex formation with MutS but not in the diffusion of MutS-MutL complex. In several experiments, the authors have formed MutS-MutL or MutS-MutL(R-E) complexes at low ionic strengths and then monitored the diffusion of proteins after buffer is exchanged to higher ionic strengths. A similar sequential injection strategy was followed to study the impact of ATP and AMP-PNP on complex formation and diffusion. The authors have also performed co-localization studies using Cy3-MutL and Cy5-MutS and obtained consistent results with previous studies. The authors also showed that MutL PCC is significant in capturing UvrD but not in interactions with MthH.

Even though the technical aspects of the study and writing of the manuscript are generally very good, I think there are two major issues in this study:

(i) It is not clear if the new findings of the study take a large enough step in the field to warrant publication in this or a more specialized journal. The experiments performed on wild type MutL are largely variants of earlier measurements and are consistent with earlier findings. There is new information gained with MutL(R-E) experiments; however, these mutations and their impact on DNA interactions of MutL were identified in earlier studies. Therefore, the new information obtained in this study is more about implications of these altered DNA interactions, reducing their originality.

(ii) The (R-E) mutations have a dramatic impact on the DNA interactions of MutL, practically eliminating them except at very low salt concentrations. The impact of this

large disturbance can be seen in many frequency and co-localization histograms where MutL(R-E) related events are completely eliminated. The disturbance is so large that even when ATP and mismatched DNA are present, MutS cannot recruit MutL(R-E) and MutL(R-E) cannot recruit UvrD under physiological ionic conditions. Introducing mutations to identify the role of a particular domain is a standard procedure but when these mutations completely eliminate a fundamental aspect that impacts many downstream physiological processes, it becomes more challenging to establish the relevance of studies.

Minor issues:

1- The authors have used different criteria for counting an event in fluorescence measurements. Sometimes 3s is considered adequate, in other cases at least 10 s or 30 s are required for counting events (Page 22-23). It is understandable that 3 s might be needed if lifetime is short; however, if 3 s is considered adequate in one case why is it not considered adequate in others?

2- Does the length of the movies impact the calculated frequencies? If a movie is recorded for 2 minutes vs. 30 minutes, are there variations in the number of molecules that might bind to DNA in 2 min vs. 30 min?

3- There seems to be significant frequency for MutL(R-E) case in Fig. 2h. Are the three frequencies observed for cases other than the first (MutL+DNA+ATP) the background levels?

4- The bin size in the histogram in Fig. 4d is significantly larger than the characteristic time (~ 40 s vs. 27 s). Given the time resolution of the experiment, the authors could consider using a smaller bin in this histogram to improve the reliability of characteristic time.

5- Page 23: Why is $0.1 \text{ } \mu\text{m}^2/\text{s}$ considered the minimum diffusion coefficient? How was this number determined? The diffusion coefficients in Fig. 2d ($\sim 0.005 \text{ } \mu\text{m}^2/\text{s}$) for MutS-MutL are about 20-fold smaller than this. Why such a high minimum value then?

6- It looks like Fig. 1e is not referred to in the text. Why are only 30 molecules used in this graph while 200-300 molecules were detected to bind to DNA in Fig. 1c?

7- The y-axis of Fig. 2d states MutS-MutL Freq but the caption says MutS-MutL(R-E) frequency. I believe the y-axis name should be corrected.

8- In Fig. 2f, how do the authors distinguish Cy3 photobleaching from dissociation of Cy3-MutL? In general, not much is mentioned about how the authors have quantified the impact of photobleaching on the dwell times they have measured. I realize that the laser exposure time is reduced in some measurements, but it would help if they state any control they have made to ensure the impact of photobleaching is negligible.

9- Why are the frequencies in Fig. 2c and 2h so different? I realize that they are different experimental configurations but is it acceptable for localization frequencies to be so different?

10- The authors use "s" or "sec"; "m" or "min". Using one version would be better,

11- Typos: Abstract: "underlining" ; Page 4, second paragraph: "includes" ; Page 5, "ensemble single molecule"; Page 6, middle paragraph: "suggests"; Page 9, 1st paragraph: "simultaneous"; Page 10, 2nd paragraph: "appears"; Page 14, last paragraph: "mechanically", also "encourages" may not be appropriate in this context; Page 18, last paragraph: "oligoes" and "neutraviding"

Point-by-Point Responses

We thank the reviewers for their valuable feedback. Please see below for a point-by-point response to the comments and concerns.

Reviewer #1

It appears that the majority of Reviewer #1 concerns centered on referencing the Fernandez-Leiro et al., (NSMB 2021) paper that appeared while the present paper was under submission. In addition, there appear to be numerous repetitive critiques. As such we apologize for the numerous repetitive answers in our effort to fully respond to Reviewer #1.

Major concerns:

1) We now include reference to Junop 2003. In addition, since the single MutL(266E) mutation has enjoyed historical context as suggested by Reviewer #1, we now include new studies comparing the intermediate effect of that mutant protein to the MutL(R162E, R266E, R316E) triple mutant protein. It is important to understand that the historical “functional” studies referred to by Reviewer #1, did not appreciate the concept of a MutL sliding clamp since it was only reported in late 2016 (Ref. 18). Thus, any data that details the role of these residues as major contributors in the formation and/or function of a MutL sliding clamp are completely novel and have not been considered by previous publications. Establishing that the MutL PCC does not confer an intrinsic DNA binding activity under physiologically significant ionic conditions but is instead utilized in a dynamic process by MutS sliding clamps to position the MutL protein for clamp loading, represents an entirely new MMR paradigm.

2) As requested by Reviewer #1 we now include reference to the Fernandez-Leiro paper (Ref. 30) throughout the manuscript. As suggested by Reviewer #1, we were unaware of these studies during the initial submission of our manuscript. We do however note that Fernandez-Leiro et al., (NSMB 2021) did not reference Liu et al., (2016, Ref. 18) or Liu et al., (2019, Ref. 20), which were the first to show the existence of MutL sliding clamps (Ref. 18) and then the mechanics of MutL sliding clamps during MMR (Ref. 20). These papers were published 5 yrs and 3 yrs prior to the Fernandez-Leiro et al., paper. As one might expect, the Fernandez-Leiro et al., paper did not consider the implications of MutL sliding clamps in their structural studies, which would have aided the MMR field in understanding the repercussions of their work as well as differences from the previous structural studies of the MutL N-terminal domain by Groothulzen et al., (eLife 2015; Ref. 31). Fernandez-Leiro et al., also did not reference London et al., (PNAS 2021, Ref. 35) that appeared during the preparation of their paper. This paper demonstrated that the human MutS homolog heterodimer, MSH2-MSH6, load the human MutL homolog heterodimer MLH1-PMS2 as sliding clamps on mismatched DNA. While this activity might have been expected for such highly conserved MMR proteins, London et al., additionally included a genetic analysis of cancer susceptibility mutations that appeared to implicate the MLH intrinsic disordered domain as essential for the thermal wrapping of MLH1-PMS2 peptides during sliding clamp formation. The notable lack of appreciation for the central role of MutL sliding clamps in MMR, clearly makes the studies described in the present manuscript both timely and valuable.

We strongly disagree with Reviewer #1 that the “mechanism and importance of the PCC are to a large extent explained by the published data that is (partially) ignored”. The relatively recent Fernandez-Leiro et al., (2021) publication clearly illustrates that the implication(s) of MutL sliding clamps have not been fully realized or generally considered by the MMR field. A similar lack of understanding could easily be articulated for the historical publications that Reviewer #1 considers as having resolved the role of the MutL PCC (see: 1) above). Prior to Nov. 2016 (Ref. 18), when majority of publications advocated by Reviewer #1 as important contributors to PCC function appeared, the MMR field as well as the larger scientific community did not know that MutL homologs formed a sliding clamp. In addition to placing those historical studies within an entirely new context, we strongly believe that the present work contributes significant additional

mechanical insights into the formation of MutL sliding clamps that are central and highly conserved components in the mechanism of MMR.

We now include new data showing the functional consequences of the MutL PCC on MutH hemimethylated GATC incision and UvrD helicase activities. The mechanics are consistent with the MutL sliding clamp tethering these downstream MMR proteins to the DNA to enhance their biochemical function during MMR. While this idea first arose in Liu et al., (2019, Ref. 20), our present manuscript provides additional supporting data for this conclusion. As recommended by Reviewer #1, we now include MutH and UvrD functional studies with *wild type* and PCC-mutant MutL sliding clamps loaded onto the mismatched DNA at very low ionic strength. *These studies now clearly demonstrated that once MutS loads a MutL sliding clamp, it plays no additional roles in downstream MMR functions.* This completely novel observation along with Ref. 18 and Ref. 20, appear to fully resolve the mechanics of *E.coli* MMR.

Finally, many of the previous “functional assays” considered as relevant by Reviewer #1 were performed under very low non-physiological ionic strength conditions or with non-circular DNA substrates where both MutS and MutL sliding clamps would rapidly dissociate from the open ends. Such non-physiological studies are intrinsically problematic and have historically led to numerous mechanical misinterpretations. The studies presented in our manuscript start with physiologically relevant conditions and then systematically alter these conditions to interrogate mechanical activities.

Other remarks:

- “The abstract mentions ‘once in a sliding clamp form, the MutL PCC aids in UvrD helicase capture but not interactions with MutH during mismatch excision’. There is however no excision data in this manuscript, so ‘during mismatch excision’ should be removed.”
We now include excision data.
- “First alinea introduction: ‘proteins mediate downstream excision events that result in the discrimination of the error containing strand’. This should be the other way around: the proteins first have to discriminate between correct and error-containing strand, only after that they incise and/or excise the error containing strand.”
The line was correct as originally conceived, assuming knowledge of the dynamic MutS and MutL sliding clamps mechanisms. However, we have changed the sentence to, “... organizing downstream MMR excision events ...”.
- “Second alinea of introduction on page 3: Together with the crystal structure (ref29) the cryoEM structure of Fernandez-Leiro from 2021 is very relevant and should be mentioned.”
Reference now included.
- “Bottom page 4: mutations of several PCC residues within this cleft resulted in impaired MMR -> include Junop et al 2003 in which the mutator phenotype of R226E is already studied, preceding the other studies.”
Junop et al.,(2003) now included.
- “Page 5 last line first result paragraph (however, the mechanical...): this is one of the sentences about known function of the PCC that should be removed/rephrased (see above).”
The sentence has been changed to, “... poorly understood”, to reflect historical data as well as the previously unknown role (or knowledge of) MutL sliding clamps.

- “Page 6, second result paragraph top: specify in text that this first experiment was performed at physiological salt, makes it easier to follow the logic without having to check in M&M.”
We now clearly indicate, “physiological ionic strength”, “reduced ionic strength”, and “very low ionic strength”, in the relevant experiments throughout the manuscript.
- “Page 7 top last sentence: this is one more of the sentences that should be rephrased because the author’s own mutator assay just confirmed that the MutL PCC has crucial physiological role instead of uncertain physiological role.”
The mutator genetic assay does not reflect any clear biophysical function, just that the component that leads to a mutator phenotype has a role somewhere in the complete MMR process.
- “Page 7 new section: First sentence mentions that a MutL activity may be disrupted, this seems to ignore that the authors just have shown in the preceding section that binding to DNA is disrupted. Rephrase more logical.”
Rephrased as suggested.
- “Page 9 last sentence upper paragraph: ‘it seems likely’ should be rephrased because the Fernandez-Leiro 2021 reference (should be included) shows exactly that.”
Referenced as suggested.
- “Page 11 at bottom: “These results suggest that the RE mutant is unable to mediate communications between mismatch recognition and strand incision/excision processes” -> acknowledge here (and in the discussion!) that this was shown before for triple RE mutant and the single R266 mutant in a more direct manner by functional analysis using incision and unwinding/excision assays in multiple studies (Junop 2003, Guarne 2004, Robertson 2006, Groothuizen 2015).”
Section re-written with significant new supporting data.
- “Page 12: “In contrast, few if any UvrD unwinding events were observed (Fig6E)”. According M&M the DNA does not contain a nick. Unwinding could therefore never occur in this setup. It could be the case that the initial MutL-UvrD interaction on its own is too weak to observe in this assay if the UvrD cannot properly engage the DNA, but that proper UvrD binding will be possible in the presence of a nick. It would therefore be interesting to repeat this assay on nicked DNA to see if UvrD in that case can actually be loaded/activated by the MutL(R-E) mutant using the low salt/high salt switch.”
The Reviewer is incorrect. Our previous work showed that the DNA used in the alluded studies contains nicks left from substrate construction (see: Ref 20). However, the section has been re-written to include new supporting data, including direct measures of UvrD helicase unwinding, that further underlining the conclusions.
- “Page 13 almost at bottom: here another of the sentences that should be rephrased: ...DNA binding activity makes the physiological role of the PCC uncertain at best.”
Sentence changed to, “... the transient and/or non-existent MutL DNA binding activity at physiological ionic strength has confounded the role of the PCC in MMR”.
- “Page 14; The first sentence is not correct. Ref 66 Schofield 2001 and Ref 67 Qiu 2015 do not report on affinities between MutS and MutL independent of DNA. All experiments were performed in the presence of DNA.”
Sentence corrected, including a prior lifetime sentence.

- “Page 14 second sentence: ‘MutS-DNA appears to position a single MutL NTD PCC in continuous contact with the DNA backbone’. Here Fernandez-Leiro has to be included because that provides direct evidence for 2 residues in the PCC to contact DNA (and the 3rd not).”

We understand that the Fernandez-Leiro et al., paper shows some but not other Arg/Lys contacts within the MutL PCC. However, we note that these are static structures as well as containing only a single MutL N-terminal domain (<30% of the entire MutL dimer). We regard it possible that other Arg/Lys residues may be utilized during dynamic MutL clamp loading or during interactions with MutH and/or UvrD. Such studies are beyond the scope of the present manuscript, but should be an interesting follow-up analysis, nonetheless.
- “Page 14 2nd paragraph 2nd sentence: rephrase to distinguish between the two following possible interpretations: The MutL sliding clamp recognizes the unwinding event created by captured UvrD, or: the MutL creates the nascent unwinding event so that UvrD can be captured (more clearly formulated in Ref20)”

Newly included data indicates the MutL PCC is required for colocalization with UvrD. Thus, the contentious paragraph has been completely re-written.
- “Page 14 2nd paragraph: the authors probably mean ‘productive UvrD binding+unwinding’ rather than capture (=initial binding) of UvrD.”

See response directly above.
- “Page 14 2nd paragraph last sentence: Include Fernandez-Leiro 2021 reference.”

Reference included in re-written paragraph.
- “Page 18: Why was MutL binding studied without magnesium? 1) Magnesium is a relevant counter ion for DNA and influences binding. 2) Is MutL actually capable of binding ATP in the absence of magnesium? How can these results be compared to later results with magnesium? What results are obtained when magnesium is included?”

Extended Data Fig. 3b from Ref 18. clearly shows that magnesium dramatically reduces the dwell time (lifetime) of MutL on the DNA. However, magnesium does not appear to alter the diffusion coefficient once on the DNA (Ref. 60). Since accurate diffusion data depends on the lifetime of the MutL-DNA interaction, we determined the diffusion coefficient of MutL alone in the absence of magnesium. All other studies were performed with magnesium. The dramatic reduction in lifetime in the presence of magnesium further underlines our conclusion that a stable MutL-DNA complex is unlikely to occur at physiological ionic conditions.
- “Page 21: M&M contains docking analysis but nowhere in the text is this referred to. Was it used for figure 7? It is better to use Fernandez-Leiro cryoEM structure to create Figure 7.”

New Fig. 8a (old Fig. 7) now uses data from Fernandez-Leiro et al., (PDB ID: 7AIB) and the docking analysis has been removed.
- “Figure 7 and Figure S1 use the wrong MutL C-terminal subunit configuration. See for example <https://doi.org/10.1016/j.jmb.2005.06.044> Please correct.”

Altered as suggested by Reviewer #1.
- “Please include information about ATP presence and concentrations in the ATPase assay description in M&M.”

ATP concentration is included throughout the Methods, including within the description of the ATPase assay.

- “It would be very relevant to include in Figure 1 a close up of DNA binding of MutL to the DNA using the cryoEM MutS-MutL on DNA structure from Fernandez-Leiro 2021.”
We have elected not to use a close-up of DNA binding from the Fernandez-Leiro et al., paper since we merely wished to indicate conserved residues and general features of the PCC. However, reference to Fernandez-Leiro et al., occurs throughout the manuscript.
- “Legend Figure 2: would be helpful to include ionic strength for all panels.”
Salt concentrations are shown in relevant panels and in the Figure Legends throughout. Placing salt concentrations within every panel appeared overly complicated and often confusing. The reader may easily access these concentrations in the Text, Figure Legend and Methods.
- “Legend Figure 4: which ionic strength?”
Salt concentrations are included in the Figure Legend and the text.
- “Legend Sup Figure 4: standard deviation on three independent experiment does not mean much because sd is a Gaussian distribution and cannot be properly described by only 3 points. It would be better to choose a different error measure that is informative. Check and correct for complete manuscript.”
Standard Error (s.e.) has been calculated for Supplementary Fig. 4 now.
- “How was labeling efficiency determined?”
Labeling efficiency is described in the Methods and was performed by spectrophotometry using the relevant extinction coefficients.
- “How was verified that labeling does not interfere with/change any function of the proteins?”
Functional equivalency of tagged proteins was determined genetically by cellular complementation by comparison of the *wild type* and tagged protein using mutator analysis. Fluorophore labeled proteins were compared to unlabeled wild type proteins for MMR activity *in vitro*. We noted no significant difference in any of these assays between tagged/labeled proteins and their corresponding *wild type*.

Minor points:

- “Please correct typo/grammar in abstract ‘mismatch nucleotides’ and ‘these studies underlining’.”
Abstract re-written
- “Last alinea page 3: rephrase ‘Dam methylation is utilized to discriminate between parental and the error-containing strand’.”
We believe the sentence structure as written is clear and flows appropriately with the preceding and succeeding sentences.
- “Correct grammar page 4 ‘where the clamp loader eventually transferring the clamp’.”
Grammar corrected.
- “The title of the first results paragraph says that the PCC is required for MMR, which is correct, but this observation is only a single sentence in this results section (and it was already shown in literature by many other studies). The main body of this result section is DNA binding by wtMutL and PCC mutant at different ionic strengths, this is the new information and I would suggest to rephrase the title of this section to reflect that.”
Please refer to reasoning contained in 2) above.
- “Page 5 first sentence second paragraph: rephrase ‘to visualize E.coli MutL PCC’ into something like ‘to study the role of the E.coli MutL PCC’.”

“Visualize” is the correct usage since those studies utilized smTIRF and single molecule tracking algorithms. Nevertheless, we have changed the sentence to, “ ... visualize activities controlled by the *E.coli* MutL PCC in real time”.

- “What is meant by ‘genetic defect’ on page 7 first line new section: The mutations from arginine to glutamate? Because they may indeed disrupt MutL activities, but better to write ‘mutations’ . Or does this refer to the mutator phenotype? Then the grammar should be changed because the phenotype is caused by changes in MutL activity rather than the other way around.”
We believe this sentence is clear as written.
- “Same paragraph: the use of ‘ensemble’ is confusing as this is a single molecule assay and ‘ensemble’ usually refers to bulk assay.”
Since these single molecule studies tracked more than one MMR component simultaneously, they are by definition ensemble studies. Nevertheless, we have deleted the word “ensemble” throughout the manuscript.
- “Same sentence: to examine MMR -> the assay does not examine the full MMR reaction, but initial steps.”
Re-written paragraph has deleted “MMR”.
- “Page 11 lower paragraph “previous studies...”: MutH does not perform excision but incision”
Section re-written to include new data that only focuses on MutH strand incision.
- “Next sentence: what is meant by “ensemble”? (see above)”
“Ensemble” removed throughout.
- “Page 13 bottom: MutL associates with MutS when it is a sliding clamp: does ‘it’ refer to MutS or MutL?”
Sentence changed for clarification.
- “Page 14 line 6: rephrase ‘MutL peptides’ to ‘regions’ or ‘domains’ of MutL”
Changed to “domains”.
- “Page 15 line 4: clarify if a protein or a nucleotide (ADP/ATP) or some other component is swapped.”
Changed to clarify “protein component”.
- “Legend Sup figure 4: Reverse order: A linear function was fit to the data.”
Legend to Supplementary Fig. 4 re-written for clarification.
- “Check all figures and correct Time (m) -> Time (minutes) (not meter).”
We believe that all “m” typos have been changed to “min”.
- “Consider replacing the 0’s (zeros) in Table S5 with ‘not relevant’.”
We have changed “0” to “ND” (not detected).5

Reviewer #2

We noted that Reviewer #2 found, “the experiments ... to be very well done”, that “this is the first evidence for a biochemical role for the PCC in MMR, which is important”, and that establishing MutS as a catalyst for the formation of MutL sliding clamps “... will significantly influence how we think about MMR mechanisms in the future”.

However, Reviewer #2 expressed concerns about the interpretation of the DNA binding role of the MutL PCC (both positive and negative). He/she also suggested an alternative model in which the MutL PCC mediated an interaction between, “both MutS and UvrD, with the MutS interaction defect [of MutL(R-E)] being suppressed at low ionic strength”. While this idea was certainly consistent with the data included in the original submission, new data presented in the revised manuscript as well as the structural studies of Fernandez-Leiro, et al., (NSMB 2021; Ref. 30) appear to strongly argue against this idea. First, the Fernandez-Leiro, et al., (2021) paper clearly shows that the MutS-MutL interaction region is completely separate from the MutL PCC region (see Ref. 30). Second, we now include new data in which we examine the ability of *wild-type* MutL and MutL(R-E) to recruit MutH or UvrD (Fig. 6 and Fig. 7). These studies directly show that MutS loads MutL(R-E) as sliding clamps at very low ionic strength similar to *wild-type* MutL at physiological ionic strength (Fig. 6e). However, unlike *wild-type* MutL, the MutL(R-E) sliding clamps are unable to recruit and enhance MutH or UvrD activities (Fig. 6f; Fig. 7e). Importantly, *wild-type* MutL loaded onto the mismatched DNA at very low ionic strength in the absence of MutS can recruit and enhance MutH or UvrD activities. Together, these observations appear inconsistent with the possibility that the MutL PCC directly facilitates the MutS-MutL interaction, and strongly supports a model in which a MutL PCC-DNA interaction is exploited by MutS to form the MutL sliding clamp and by MutH and UvrD during their respective DNA transactions.

To ameliorate Reviewer #2 *minor note*, we have expanded the Figure Legends to increase their self-explanatory nature, as well as enhanced the kymograph exposures for clarity.

Specific comments:

- “P5 and Figure 1a. I think figure 1a would be more informative if it also showed the basic patch in a MutL N-terminal dimer so that one can better see the positioning of the basic patch. In addition, the positions of the amino acid substitutions used should be shown.”
We now show the basic patch of the MutL N-terminal dimer in Fig 1a, as well as its positioning during MutL sliding clamp loading in Fig. 8a.
- “P6 and Supp Figure 2. It would be helpful if the legend for panels d, e and f contained some additional information about each experiment (what was added, ionic strength, etc) and also what defines the starting position (presumably first visible fluorescence). In general, all of the legends should be looked at from this perspective.”
Additional information included in the legend to Supplementary Fig. 2d-f, including what constitutes a “starting position”.
- “P6, L11. DNA are either”
Grammar corrected
- “P6, L16-18. Do the authors really have data that allows concluding ion shielding and rotational coupled diffusion?”
These data are largely contained in Ref. 18 (with associated references).
- “P7, L1. An elevated frequency of ...”
Sentence changed as requested.
- “P7, L6, 7. The genetics establish a physiological role. What they don't establish a mechanism underlying this role.”
Sentence re-written for clarification.
- “P7, L12. Found a small difference, not very little difference.”
Sentence changed as suggested.

- “P7, L22. Aren't the authors substituting MutL (R-E) for wild-type MutL?”
Paragraph re-crafted for clarification.
- “Figure 2e legend. What ionic strengths were used in the two experiments (are they the same or different given the lack of recruitment of R-E at physiological ionic strength).”
Salt concentration and ionic strength are now included in the Fig. 2e and the Figure Legend.
- “P9, L8-11. It would be nice to have the 2 sites indicated on a structure to support this statement. Possible Supp Figure 1.”
Contacts are now shown in Fig. 8a (from Fernandez-Leiro, et al., 2021; Ref. 30).
- “P13, L9. Dynamic motion on DNA. Aren't there some SPR references that should be cited?”
Additional SPR reference now included.
- “P14, L21. Mechanical, not mechanically.”
Sections entirely re-crafted based on new data.
- “Legend to Figure 7b. This legend is truly inadequate. Most of the proteins in this figure panel are not even identified.”
Fig. 7b (Now Fig. 8b) includes additional protein legend, short descriptions of mechanics, and enhanced figure legend with reference.

Reviewer #3

Reviewer #3 noted that even though, “the technical aspects of the study and writing of the manuscript (were) generally good”, he/she expressed “two major issues in this study”:

The first questioned whether the “new findings take a large enough step in the field to warrant publication (in Nature Communications) ...”, since the MutL PCC mutations “... and their impact on DNA interactions of MutL were identified in earlier studies”. In response, it is worth emphasizing again that *there is no “DNA binding” by MutL at physiological ionic strength*. With this central observation the biophysical role of the MutL PCC residues immediately became mysterious. Moreover, many of the implied historical functions during MMR were questionable at the very least and at most, completely wrong.

As explained under 2) for Reviewer #1 above, it seems quite clear that the importance of MutL sliding clamps in the mechanism of MMR has not been fully or generally appreciated by the MMR field. The studies presented in our manuscript clearly connected the MutL PCC to MutL sliding clamp formation. We show that MutS functions *with* DNA as a MutL clamp loader. The mechanics of the MutS-DNA clamp loader activity are dramatically different from the canonical replication clamp loaders that attach β -clamp or PCNA to a replication fork. New studies included in the resubmission demonstrate how MutL sliding clamps mediate downstream MutH incision and UvrD unwinding activities during MMR, both of which also appear to utilize the MutL PCC. Since MutS-dependent MutL clamp loading, MutH hemi-methylated GATC incision, and UvrD helicase displacement of the mismatch-containing strand are all dynamic processes on the msec timescale, our manuscript presents a completely new paradigm for the role of the MutL PCC in MMR mechanics.

The second issue surrounds the suggestion by Reviewer #3 that mutations, such as MutL(R-E), which “completely eliminate a fundamental aspect that impacts many downstream physiological processes ...”, makes it “... more challenging to establish the relevance of studies”. Our revised

manuscript now includes data where a single MutL PCC residue [MutL(R266E)] has been altered. Compared to the MutL(R-E) triple mutation, the MutL(R266E) proteins displays an intermediate genetic defect (mutation frequency) and remains active at intermediate ionic strength where MutL(R-E) is completely defective (see: Fig. 1c; Fig. 3d). Ultimately, these studies strongly correlate ionic strength dependent loss of DNA binding activity with a corresponding ionic strength dependent loss of MutL sliding clamp formation (both MutS-dependent and MutS independent; Fig. 3d). Taken as a whole these observations further support our conclusion that MutS acts as a catalyst for MutL clamp loading by dynamic positioning the MutL PCC on the DNA. As a general rule, the more amenable the MutL PCC is to positioned DNA association, the more likely MutS can catalyze the formation of a MutL sliding clamp. Rarely are biochemical experiments black-and-white. But to our knowledge, these studies represent the first multi-factorial quantitative analysis of MutL PCC function and its connection to MutS-catalyzed formation of a MutL sliding clamp.

Minor issues:

1. “The authors have used different criteria for counting an event in fluorescence measurements. Sometimes 3s is considered adequate, in other cases at least 10 s or 30 s are required for counting events (Page 22-23). It is understandable that 3 s might be needed if lifetime is short; however, if 3 s is considered adequate in one case why it is not considered adequate in others?”

Based on our current and previous studies (Ref. 18), MutL displays a lifetime of ~8 sec on DNA, the MutS-MutL complex displays a lifetime of ~30 sec, and the MutL sliding clamp has a lifetime of ~14 min. Thus, to examine these various molecular forms, movies were recorded at different frame rates (100 – 6000 msec) and exposure times (100 – 300 msec). In addition, we only evaluated molecules that were clearly diffusing along the DNA (> 5 frames). Therefore, we used different cut-offs for different analysis. We now include these details in the methods.

2. “Does the length of the movies impact the calculated frequencies? If a movie is recorded for 2 minutes vs. 30 minutes, are there variations in the number of molecules that might bind to DNA in 2 min vs. 30 min?”

The frequency of molecular events increases when movies were recorded for longer times. This is because the numbers of events increased, while numbers of DNA molecules do not. However, these time-dependent variations equally influence *wild type* and mutant proteins. For individual studies where frequencies are compared, the movie length always remains constant. We now include these details in the methods.

3. “There seems to be significant frequency for MutL(R-E) case in Fig. 2h. Are the three frequencies observed for cases other than the first (MutL+DNA+ATP) the background levels?”

Reviewer #3 is correct in his/her interpretation of background levels. These background signals likely represent non-specific or random co-localizations of MutS and MutL proteins on the surface. See for example the -ATP control where there should be no functional interactions between MutS and MutL.

4. “The bin size in the histogram in Fig. 4d is significantly larger than the characteristic time (~40 s vs. 27 s). Given the time resolution of the experiment, the authors could consider using a smaller bin in this histogram to improve the reliability of characteristic time.”

We thank Reviewer #3 for this suggestion. We have now used a smaller bin size for Fig. 4d.

5. “Page 23: Why is 0.1 $\mu\text{m}^2/\text{s}$ considered the minimum diffusion coefficient? How was this number determined? The diffusion coefficients in Fig. 2d (~0.005 $\mu\text{m}^2/\text{s}$) for MutS-MutL are about 20-fold smaller than this. Why such a high minimum value then?”

Our previous studies have demonstrated that diffusion coefficients may be distributed across 2 magnitudes (Ref. 18). We select $0.1 \mu\text{m}^2\text{s}^{-1}$ as a cut-off to clearly discriminate a MutL sliding clamp ($0.88 \pm 0.39 \mu\text{m}^2\text{s}^{-1}$) from the MutS-MutL complex ($0.004 \pm 0.002 \mu\text{m}^2\text{s}^{-1}$). See Ref. 18, Extended Data Table 4.

6. “It looks like Fig. 1e is not referred to in the text. Why are only 30 molecules used in this graph while 200-300 molecules were detected to bind to DNA in Fig. 1c?”

We have now referred to Fig. 1e in the text. Single molecule analysis of diffusion coefficients is quite time consuming and requires at least 50 frames to evaluate and calculate the square of the distance from the previous location in each frame. For Fig. 1e the lifetimes must be >10 s. In contrast, molecules with a minimum lifetime of 3 s could be easily counted as binding events in Fig. 1c.

7. “The y-axis of Fig. 2d states MutS-MutL Freq but the caption says MutS-MutL(R-E) frequency. I believe the y-axis name should be corrected.”

The name has been corrected.

8. “In Fig. 2f, how do the authors distinguish Cy3 photobleaching from dissociation of Cy3-MutL? In general, not much is mentioned about how the authors have quantified the impact of photobleaching on the dwell times they have measured. I realize that the laser exposure time is reduced in some measurements, but it would help if they state any control they have made to ensure the impact of photobleaching is negligible.”

We thank Reviewer #3 for this suggestion. We have performed a photobleaching experiment to ensure the impact is negligible in our single-molecule imaging system (Please see Response Fig. 1 as below).

9. “Why are the frequencies in Fig. 2c and 2h so different? I realize that they are different experimental configurations but is it acceptable for localization frequencies to be so different?”

The frequencies shown in Fig. 2c and Fig. 2h were developed from completely different experiments that utilized different starting substrates. For Fig. 2c, we immobilized DNA on the flow cell surface and identified MutS-MutL complexes diffusing along the DNA. The MutS-MutL complex was observed at a frequency of 0.6 per DNA molecule (60% of the DNA molecules displayed a colocalized MutS-MutL complex). In contrast, for Fig. 2h we immobilized a single Cy5-MutS on surface and visualized co-localized of Cy3-MutL particles. We found that $\sim 10\%$ of the Cy5-MutS displayed a co-localization event with

Cy3-MutL in the presence of mismatch DNA. There are at least two reasons for the difference in MutS-MutL complex frequency: 1) For data in Fig. 2c, multiple MutS sliding clamps could be loaded on single DNA molecule (see: Fig. 2a), which would increase the frequency of co-localization events; and 2) The efficiency of MutS sliding clamp formation (a minimum requirement for the studies in Fig. 2c) is likely to be lower when the protein is bound to the flow cell surface (Fig. 2h), leading to a reduced frequency of MutS-MutL complex. Thus, MutS proteins that do not adopt the sliding clamp configuration (such as a searching MutS) would not affect the frequency in Fig. 2c but would reduce the frequency in Fig. 2h.

10. "The authors use "s" or "sec"; "m" or "min". Using one version would be better."
Corrected.

11. "Typos: Abstract: "underlining" ; Page 4, second paragraph: "includes" ; Page 5, "ensemble single molecule"; Page 6, middle paragraph: "suggests"; Page 9, 1st paragraph: "simultaneous"; Page 10, 2nd paragraph: "appears"; Page 14, last paragraph: "mechanically", also "encourages" may not be appropriate in this context; Page 18, last paragraph: "oligoes" and "neutraiding"."
Typos corrected

Reviewer #1 (Remarks to the Author):

I have read the revised version of Yang et al. with great interest and in my opinion the manuscript has improved. The new data showing the functional consequences of the PCC mutants is valuable. However, the new version does not fully address all my original concerns, and the new data raise issues that could be more elaborately discussed.

- The missing Fernandez-Leiro reference, together with the missing Junop 2003 reference, were indeed among my major points of concern, but easily fixable, as has been done by the authors.
- However, that leaves my main point of concern, that the new findings in this study come down to the involvement of the PCC residues in MutL clamp formation, while the role of the PCC in DNA binding, MutH activation, UvrD activation and overall repair have already been published and that this is not properly discussed in the manuscript. Also see Reviewer 3 ("if new findings take a large enough step in the field to warrant publication in Nature Comm") and my last sentence above the line 'other remarks': "Especially in the discussion the reported findings about MutH and UvrD binding have to be put in context of the published functional assays". In the second half of the discussion, about DNA binding, MutH and UvrD activation, no historic MMR studies are being referenced; not in the original manuscript and not in this revised version.
 - o In the rebuttal the authors correctly mention that the historical studies, and even recent work such as Fernandez-Leiro, did not consider the fact and importance of MutL sliding clamp formation. They also mention that with this work they place these historical studies within an entirely new concept, but in my opinion they actually do not, as the historic papers are not discussed (see above).
 - o In the rebuttal the authors dismiss many of the previously published studies because they have been performed under non-physiological conditions. However, this is not the case for all studies, furthermore it does not mean all results are useless (the authors themselves also use non-physiological conditions in specific cases), and most importantly, this is not explained in the discussion, so that interested readers from the field will be unable to carefully weigh the new insights versus the old. These considerations should be discussed.
 - o The new functional data with MutH nicking and UvrD unwinding activity assays give interesting results. In fact, they show that the PCC is not only involved in MutL sliding clamp formation, but also in MutH activation (not binding) and UvrD binding and activity. These activities were already addressed in the historical studies and thus the need for a proper comparison of the new data with the old assays is even more relevant in this manuscript version.
- It is intriguing that MutH binding is unchanged but GATC site incision is not possible with the PCC mutant. Can the authors elaborate more in the discussion about possible mechanism?
- The abstract has been rewritten but there are 2 sentences that should be reconsidered:
 - o 'Classical models envision static MutS-MutL complexes': There are also classical models that envision mobile complexes (for example the molecular switch model). I think a sentence about complexes but without references to models, might work better here.
 - o As the authors mention in the discussion, the diffusion-mediated wrapping of the remaining MutL domains around the DNA is a hypothesis/likely scenario, but there is no experimental evidence yet. However, the current statement in the abstract makes it sound like a fact. This would be better rephrased.
- Something the authors could consider (I realise I did not bring this up in first version); The manuscript has focus on role of MutL-PCC, while the title now has focus on MutS.
- Page 7: The first sentence in paragraph "DNA is essential for the MutS-MutL interaction" has been rephrased but still is incorrect in grammar. The increased frequency of mutations (ie colonies) found (the genetic defect) is a consequence. The amino acid changes in R266E and RE can disrupt activity, that will result in the mutator phenotype.

Reviewer #2 (Remarks to the Author):

Yang et al NCOMMS-21-21036B

This is a revision of a paper that I previously reviewed. The authors have addressed my concerns. As part of the revision process, they have also added additional data which I found helpful.

The paper describes a detailed study, mostly using single molecule methods, analysing loading of different MMR proteins onto DNA. These loading reactions are primarily initiated by MutS loading at mismatches followed by recruitment of MutL and other proteins. The methods for analysis are interesting and will be useful to others. In my view, there are two key results. One is the identification of a role for the DNA interacting domain of MutL at physiological ionic strength. This is interesting because this region of MutL doesn't actually interact with DNA at physiological ionic strength. The other is that MutL can be loaded onto DNA at low ionic strength, form clamps and facilitate downstream reactions with MutS. This is particularly interesting because it both demonstrates a role for the MutL DNA binding region and also solidifies the view that at least in *E. coli* MutS is primarily a MutL loading factor and may not otherwise participate in downstream reactions. To me, this is thought provoking.

Minor comments

Page 7, L5. MutL mutant plasmids

Page 7, L19. Don't the authors mean to say "...since MutL DNA binding was eliminated at ~80 mM ionic strength, which is well below physiological ionic strength."

Page 7, L21. The line is not clear. Don't the authors mean to say that the two different MutL defects cause either a mutator phenotype or increased accumulation of mutations?

Page 10, L11. What does significant mean? Don't the authors mean to say they are formed at a detectable frequency? And I would find it useful if the authors actually put the frequency seen in the text rather than just depending on the heat map plus a guess at the frequency represented by the shade of red.

Figure 2, 3 and elsewhere. I would find it useful if the authors would label at least some of the MutS and MutL protein depictions in each figure "MutL" or "MutS" rather than depending on the reader to remember the color coding. A minor issue.

Page 12, L2. Should the authors state that release is mediated by hydrolysis that allows the N-termini to dissociate opening the ring as aren't both ends of the DNA blocked.

Figure 7a and legend. Is there MutS and a mismatch in these reactions? Isn't MutS loading required to load MutL under physiological ionic strength? Some additional detail is needed in the legend.

Page 15, L16 At physiological ionic strength. Correct?

Reviewer #3 (Remarks to the Author):

The authors have made significant changes to the manuscript, have included new data, updated some figures and schematics, and clarified/corrected parts of the writing that were highlighted in referee reports. They have also included new data with MutL (R266E) which clarified one of my major concerns.

My other major concern was whether this study presents a significant enough advance in our understanding to be published in *Nat. Comm.* A similar issue appears to be brought up by Reviewer 1 as well, and the authors have provided an extensive response to illustrate the novel aspects of the current study. However, given the recent literature (particularly the 2016 and 2019 papers by the recent authors-Liu et al.), the advancement in this paper still appears to be relatively modest. For example, the authors strongly emphasized the observation that there is no DNA binding by MutL at physiological salt (at least within the time resolution of the smTIRF studies); however, there are many studies, including those by the authors, that have established that MutL is recruited to DNA by MutS. Given this known mechanism, the significance of no DNA binding by MutL at physiological salt is not as prominent. Similarly, the authors dismiss many of the studies before 2016 since they were done at low ionic conditions. However, I am not convinced

that such studies could as easily be dismissed since the observed impact of PCC domain, and particularly of the three identified aminoacids, after MutL is recruited to DNA are still important and seem to be in line with the recent structural studies that show at least two of the three aminoacids are in contact (close proximity) of the DNA in the static images. As the authors mention, the third also interacts with DNA during the dynamics of the complex.

Many of the ideas presented in this paper were also studied and mentioned in their earlier work, even if they came after 2016. For example, the Liu et al. 2019 paper showed MutL sliding clamp captures UvrD near ssDNA break and the two working together to displace ssDNA between GATC adjacent incisions. This is not to suggest that a complex mechanism such as MMR can be understood in a single study. However, several such incremental advances have made it more challenging to assess whether the current study provides a large enough advancement.

In short, I think this is an important study that will enrich the literature on the role of MutL on MMR and particularly the role of MutL PCC on MutL sliding clamp formation and downstream activities of MutH and UvrD. However, it is not clear if the authors make a strong enough case to justify publication in this journal.

We thank the reviewers for their thorough assessment and believe that the critiques have been fully addressed. Taken as a whole, our studies appear to provide a unique and nearly complete understanding of the biochemical progressions that result in MMR.

General

*For Reviewers #1 and #3 we wish to stress that the MutL protein does not stably bind to DNA at physiological salt (<100 msec). This extremely short time interval virtually eliminates canonical MMR models where solitary MutL DNA binding plays any substantial role. To be sure, historical and more recent structural studies (Ref. 30, 49, 61) showed that mutation of Arg/Lys residues located in a MutL N-terminal domain positively charged cleft (PCC) reduced or eliminated a DNA binding activity, which could only be detected at low ionic strength. These same MutL PCC mutations resulted in a mutator phenotype *in vivo* solidifying a role in MMR. Companion studies showed that PCC mutant MutL proteins were less efficient than *wild type* MutL at stimulating UvrD helicase activity *in vitro* as well as less efficient in a “complete” mismatch-MutS-MutL-dependent MutH endonuclease strand scission assay – both initially performed under low ionic strength conditions (Ref. 49, 63). These observations seemed to underscore MMR models where a MutL DNA binding activity was directly connected to downstream excision processes.*

However, once we found that the MutL PCC is an essential component employed by MutS with DNA to load MutL sliding clamps, then all the historical studies connecting the PCC to MutH or UvrD activities could have been directly linked to the efficiency of MutL sliding clamp formation. Our studies have gone the extra step to clearly delineate multiple distinct MutL PCC engagement practices that included MutS-dependent MutL sliding clamp formation as well as MutH endonuclease and UvrD helicase progressions. These observations are worth repeating. *We clearly demonstrated the multiplex functional organization of an otherwise undetectable MutL PCC DNA binding activity, by three distinctly separate MMR protein complexes with unmistakably different DNA metabolic activities.* To our knowledge this type of multiplex idiosyncratic operational management has never been shown for any biochemical reaction(s). The fact that generally underappreciated MutL sliding clamp operations play a central role in this unusual mechanics, makes the observations presented in our manuscript crucial for understanding the mechanism of MMR.

Reviewer #1

See *General* above for considerations regarding previously published MutL DNA binding studies. We believe that the revised Discussion places the historical studies within the entirely new mechanics of MutL sliding clamp formation and operations. Perhaps more importantly, we demonstrate that the only role for MutS is as a clamp-loader for MutL sliding clamps. This is an entirely new concept that should enrich discussions regarding the similarities and difference between *E.coli* MMR and MMR mechanisms utilized by organisms outside γ -proteobacteria.

- To our knowledge the only biochemical studies that examined the MutL PCC at physiological ionic strength was contained in Ref 30. These studies described a “complete” mismatch-MutS-MutL-dependent MutH endonuclease strand scission assay, which would have unknowingly employed MutL sliding clamps. We now include extensive discussions of salt-dependent sliding clamp loading, with and without MutS. These observations provide significant new insights into these historical studies by clearly connecting them to the MutL PCC during MutL sliding clamp operations.
- We have endeavored to frequently compare our present studies with previous observations throughout the manuscript and in particular within the context of the Discussion. This includes prior MutH nicking and UvrD unwinding studies. We do not deny that these were thought provoking historical observations. But these previous studies exposed significant limitations since they did not probe the interactive progressions of MutS, MutL, MutH or UvrD with DNA at physiological ionic strength, nor

did these studies anticipate the formation of a MutL sliding clamp or the prospective mechanical functions of a MutL sliding clamp in the multi-step reactions that leads to MMR. With the recognition of MutL sliding clamps, the possible role(s) for the MutL PCC in MMR were considerably expanded to encompass crucial interaction(s) between MutS and MutL, the material formation of the MutL sliding clamp and/or the distinctive collaborations between the MutL sliding clamp with MutH or UvrD. The studies contained in our manuscript answer a great many of these unsettled questions and provide a solid new framework for how MMR works *in vivo*.

- In both the text and discussion, we point out that the association of MutH with MutL sliding clamps is unaffected by Arg/Lys mutations of the MutL PCC. In contrast, the ability of MutH to incise hemi-methylated GATC sites is significantly compromised by mutations of the MutL PCC. We regard it likely that the MutL PCC is (at least partially) engaged when MutH binds to the MutL sliding clamp. Such an engagement should help MutH search for hemi-methylated GATC sites. As an aside, MutH is structurally related to the Sau3A restriction endonuclease that recognizes GATC sites independent of Dam-methylation. Importantly, Sau3A normally functions as a homodimer. Yet, MutH functions as a monomer with abysmal DNA binding or nicking activity on GATC sites regardless of methylation status. One might predict that the engagement of the MutL PCC by the MutL-MutH complex will encourage continuous contact with the DNA backbone and associated rotation-coupled diffusion (see: Ref. 19). Such close DNA contacts could in theory substantially enhance the MutH search for hemi-methylated GATC sites. These studies are currently underway but are certainly beyond the scope of the present manuscript. We have now included a sentence in the Discussion suggesting a clear difference between MutL-MutH complex formation on the DNA and the MutL-MutH search for hemi-methylated GATC sites.
- We have removed “classical” from the abstract as suggested by Reviewer #1.
- We have rephrased the offending sentence to, “..., *likely* enabling diffusion-mediated wrapping of the remaining MutL domains around the DNA”.
- We believe that most novel concept contained in the current manuscript is that MutS *uniquely* acts as a MutL clamp loader in MMR. Moreover, we provide plausible mechanics for this process that involves a novel role for the MutL PCC, which we find is employed by the MutS sliding clamp *with DNA* in physiological ionic conditions. Thus, we believe the original title adequately expresses the contents of the paper.
- We believe the sentence in the revised manuscript is now correct.

Reviewer #2

We have made the requested corrections to address all of the minor comments expressed by Reviewer #2.

Reviewer #3

Please see *General* above as well as answers to specific comments from Reviewer #1

- We have included as sentence regarding the time resolution of our smTIRF studies (100 msec), suggesting that we cannot detect any MutL DNA binding above this very short frame rate. This is an important point. However, since single molecule detection of DNA binding generally fits a single exponential decay, the fact that we observed no binding at a 100 msec frame rate suggests that any MutL DNA binding activity must be significantly

faster than this interval. Such a brief interaction would seem to be largely insignificant for biological reactions.

- We have substantially recrafted the Discussion to clearly show that our studies provide more than incremental advances to the MutL literature, the MMR field, and the study of protein-DNA complexes.

Reviewer #1 (Remarks to the Author):

This revised version of the manuscript is much improved, the more elaborate discussion about salt-dependent DNA-binding puts the new data in a historical perspective that is very useful and will help the field forward. My concerns have been addressed, except for the first sentence in paragraph "DNA is essential..." on page 7 which is still not correct (The increase in mutations is a consequence, cannot be a cause). I suggest to rephrase like this: The increased frequency of mutations found with [...] could be the result of the mutations disrupting one or more of the known MutL activities during MMR.

Reviewer #1

“This revised version of the manuscript is much improved, the more elaborate discussion about salt-dependent DNA-binding puts the new data in a historical perspective that is very useful and will help the field forward. My concerns have been addresses, except for the first sentence in paragraph "DNA is essential..." on page 7 which is still not correct (The increase in mutations is a consequence, cannot be a cause). I suggest to rephrase like this: The increased frequency of mutations found with [...] could be the result of the mutations disrupting one or more of the known MutL activities during MMR.”

We have revised the sentence for clarity and without the repetition of “mutation” that might confuse the reader.